# ONLINE-TO-OFFLINE RL FOR AGENT ALIGNMENT

**Xu Liu[1,2], Haobo Fu[2], Stefano V. Albrecht[3], Qiang Fu[2], Shuai Li[1]**[*]
Shanghai Jiao Tong University[1], Tencent[2], University of Edinburgh[3]
{liu_skywalker,shuaili8}@sjtu.edu.cn
{haobofu,leonfu}@tencent.com, ste.albre@gmail.com

## ABSTRACT

Reinforcement learning (RL) has shown remarkable success in training agents to achieve high-performing policies, particularly in domains like Game AI where simulation environments enable efficient interactions. However, despite their success in maximizing these returns, such online-trained policies often fail to align with human preferences concerning actions, styles, and values. The challenge lies in efficiently adapting these online-trained policies to align with human preferences, given the scarcity and high cost of collecting human behavior data. In this work, we formalize the problem as *online-to-offline* RL and propose ALIGNment of Game AI to Preferences (ALIGN-GAP), an innovative approach for the alignment of well-trained game agents to human preferences. Our method features a carefully designed reward model that encodes human preferences from limited offline data and incorporates curriculum-based preference learning to align RL agents with targeted human preferences. Experiments across diverse environments and preference types demonstrate the performance of ALIGN-GAP, achieving effective alignment with human preferences.

## 1 INTRODUCTION

Modern reinforcement learning (RL) has achieved significant successes in optimizing strategies across diverse environments, driven by innovative algorithmic designs and the integration of deep learning techniques (Li, 2017; Wang et al., 2022). Despite the high sample complexity typically associated with RL training, practical applications can mitigate online training costs and develop effective policies through approaches such as reward shaping (Schäfer et al., 2022) and pre-training on substantial offline datasets (Levine et al., 2020; Prudencio et al., 2023; McInroe et al., 2024; Andres et al., 2024). In the realm of Game AI, the availability of cost-effective game engines facilitates efficient interactions, enabling RL agents to achieve high returns under carefully designed reward functions (Shao et al., 2018; Fan et al., 2020). However, integrating these agents into real-world applications often uncovers a misalignment between the agents' behaviors, styles, and values and those of humans, particularly given the varied value orientations of different individuals. As an example illustrated in Figure 1, an online-trained agent learns a behavioral policy based on a given reward function. However, diverse values can exist for human game players, e.g., a cautious beginner, a speed-runner focused on completion time, or an achievement hunter aiming for in-game accomplishments. This diversity in values could lead to a variety of play styles and behaviors among different player groups. This discrepancy can make the agents appear unnatural from a human perspective, necessitating alignment of agents with specific human preferences.

To align agents with human preferences in aspects such as human-like behaviors and action styles, it is essential to encode human values and preferences from behavior data, which may be significantly off-policy for the online-trained agents (Ziegler et al., 2019). The high cost and scarcity of collecting human behavior data further hinder the effectiveness of direct fine-tuning processes. Furthermore, the variety of behavioral preferences, styles, and values among humans necessitates alignment with specific groups or even individual players, intensifying the challenges associated with data collection (Gabriel, 2020). The inherent sample complexity of RL algorithms presents significant challenges in fine-tuning. Besides, the potential suboptimality of human-preferred behaviors, when assessed through the agents' value estimations, further complicates the process (Nguyen-Tang et al.,

---

[*]Corresponding author.

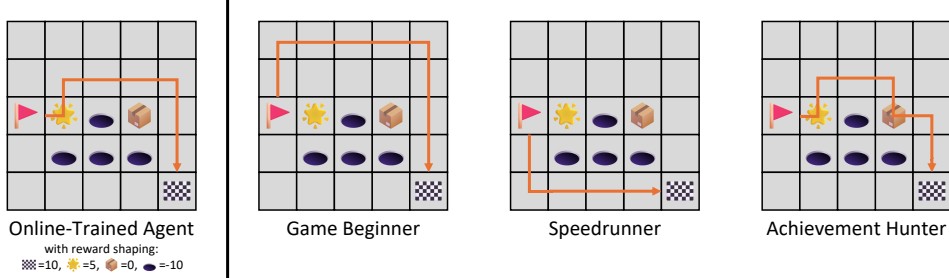

**Slippery run:** Navigate a slippery surface from starting flag ▶ to reach the finish line ▩ without falling into pits ⬭ . Inexperienced players may sometimes slide unpredictably. Collect stars ☀ and open boxes 🎁 (which have a 50/50 chance of containing a star) for bonus achievements. Reaching the finish line or falling into a pit ends the game.

Figure 1: An example for agent behavior (left) and diverse human behaviors (right) in a grid-based game. A beginner prioritizes avoiding pitfalls; a speed-runner optimizes for quick completion; an achievement hunter seeks to collect stars and open boxes for possible collectable items.

2021). Additionally, the shift between the state-action distributions of human-preferred styles and those of RL agents can lead to "unlearning" phenomena during fine-tuning, whereby agents lose their previously acquired understanding of environmental dynamics (Kumar et al., 2020; Kostrikov et al., 2021; Nakamoto et al., 2024). To address these challenges, we conceptualize the issue as an *online-to-offline* RL problem. Here, "online" refers to the process of achieving optimal or near-optimal policies through extensive online RL training or imitation of high-return trajectories, as evaluated by reinforcement learning rewards from the environment. The "offline" component involves aligning these well-trained policies with human preferences using offline behavior data, which can often be scarce. This framework contrasts with offline-to-online RL, where key distinctions can be summarized under two main categories:

- **Assumptions regarding offline data:** Offline-to-online RL typically utilizes abundant but potentially suboptimal offline data, which provides extensive coverage of environmental dynamics (Kumar et al., 2020; Kostrikov et al., 2021). Online-to-offline RL, however, focuses on offline preference data reflecting specific human behaviors or values. This data is often limited and costly to collect, which may hinder its utility for fine-tuning RL policies.
- **Optimization objectives:** The goal of offline-to-online RL is to enhance online learning by utilizing offline pre-training to address challenges in sample complexity (Zhang et al., 2023; Hansen-Estruch et al., 2023; McInroe et al., 2024). In contrast, online-to-offline RL aims to align online-trained agents with human preferences encoded in offline behavior data, prioritizing the integration of human values into the learning process.

In this work, we introduce **ALIGN**ment of **G**ame **AI** to **P**references (**ALIGN-GAP**), aimed at aligning RL agents with human styles and values within the Game AI context. Our method comprises two stages: first, inspired by techniques from large language model alignment, we extract human preferences from offline human data by training a transformer-structured reward model. This model is trained with pair-wise sampling of state-action (sub-)trajectories from both online-trained agents and human data. Subsequently, to bridge the gap between the objectives of online learning and aligning with human preferences, we introduce a calibrated preference curriculum learning strategy. This approach begins by calibrating the reward model scores to identify behaviors that align with human preferences but are not yet learned by the agent. It then gradually shifts the agent's objectives, moving from maximizing environmental returns to prioritizing human preferences as encoded by the reward model. ALIGN-GAP achieves strong performance on D4RL locomotion and Atari games, outperforming baseline fine-tuning methods in alignment on 17 of 21 tasks.

## 2 RELATED WORK

**Online RL with offline data.** Prior works have explored various strategies to utilize previously collected offline datasets to accelerate the training of online RL algorithms. These approaches can

generally be divided into two main categories: offline-to-online RL and online RL with offline data. Offline-to-online RL involves pre-training a policy and a value function using offline data (Kumar et al., 2020; Kostrikov et al., 2021; Agarwal et al., 2020; Nakamoto et al., 2024; Zhang et al., 2023), which allows for effective utilization of abundant yet sub-optimal offline datasets to establish well-initialized policies. This reduces the need for extensive online exploration, thereby improving sample complexity and overall efficiency (Prudencio et al., 2023; Levine et al., 2020; McInroe et al., 2024; Zhong et al., 2022). To address the distributional shift between the offline dataset and online interactions, several techniques such as balanced sampling (Lee et al., 2022; Guo et al., 2023), adaptive conservatism (Kostrikov et al., 2021; Nakamoto et al., 2024), and actor-critic alignment (Yu & Zhang, 2023; Wang et al., 2024b) have been proposed to enhance stability and performance. In contrast, online RL with offline data involves training an agent from scratch on top of a replay buffer filled with online data and offline data (Song et al., 2022), mixed with approaches such as imitation learning (Andres et al., 2024) and symmetric sampling (Ball et al., 2023). In our work, we seek to extend beyond improving agent performance by addressing how to align well-trained agents with human preferences using a limited amount of human offline data. Here, the challenge shifts from maximizing environmental returns to aligning agents with specific human preferences, where offline data may be of high quality but insufficient for extensive RL fine-tuning.

**Learning from human preferences.** With the evolution of large language models (LLMs), aligning AI capabilities with human values has become a focal point of research. In the domain of RL, modern algorithms enable agents to learn effective strategies through extensive interaction with complex environments (Arulkumaran et al., 2017; Schulman et al., 2017; Haarnoja et al., 2018). A primary step for aligning these capable agents with human values involves representing and encoding human preferences, especially from pre-collected human data. Inverse reinforcement learning has been used to infer reward functions that could explain the behavioral patterns hidden within the data (Eysenbach et al., 2020; Xue et al., 2021), although this often requires a significant amount of demonstration data (Arora & Doshi, 2021; Adams et al., 2022). A promising alternative is the development of reward models that encode human preferences with transformer-based architectures (Guan et al., 2022; Dong et al., 2023; Faal et al., 2023; Kim et al., 2023), which can help capture long contextual relationships and dependencies effectively through pair-wise sampling and training (Ouyang et al., 2022; Wang et al., 2023). These preference signals have found successful applications in AI model alignment, particularly in LLM and vision language model alignment (Guan et al., 2022; Liu et al., 2023; Wang et al., 2024b).

**Aligning agents with humans.** Popular methods for aligning fundamentally capable base models with human preferences include using reward models and proximal policy optimization (PPO) in reinforcement learning from human feedback (RLHF) framework (Achiam et al., 2023; Wang et al., 2024c;a). By formalizing the alignment challenge as a bandit problem, initiatives like InstructGPT improved its alignment capability on issues such as hallucination and toxicity (Ouyang et al., 2022). RLHF has been adopted in various subsequent LLMs (Zheng et al., 2023; Wang et al., 2024a; Yang et al., 2024). Moreover, directly using offline human data through approaches like direct policy optimization (DPO) allows for supervised fine-tuning that aligns base models more closely with human expectations of being helpful and harmless (Rafailov et al., 2024; Xu et al., 2024). Further explorations have relaxed the form of offline data (Ethayarajh et al., 2024) and integrated reward signals to improve alignment (Zhong et al., 2024). In the field of reinforcement learning, there has been an exploration of aligning agents with humans in domains such as autonomous driving and gaming (Wurman et al., 2022; Dong et al., 2023). However, existing approaches often require hand-engineering attributes for specific environments and struggle to provide a unified formulation across different domains. In this work, we aim to identify the challenges and bottlenecks in aligning RL agents with humans through an *online-to-offline* RL framework and propose an alignment approach under this setting to establish a general strategy for agent-human alignment.

## 3 PRELIMINARIES

### 3.1 REINFORCEMENT LEARNING

Reinforcement learning (RL) involves training an agent to make decisions by interacting with an environment in a way that maximizes cumulative rewards over time. This is formalized within

the framework of a Markov Decision Process (MDP) (Puterman, 1990), represented as $\mathcal{M} = (\mathcal{S}, \mathcal{A}, P, r, \rho, \gamma)$, where $\mathcal{S}, \mathcal{A}$ denote the state and action spaces, $P(s'|s, a)$ is the state transition probability, $r(s, a)$ represents the reward function, $\rho(s)$ is the initial state distribution, and $\gamma \in (0, 1)$ is the discount factor which prioritizes immediate rewards over future ones.

The objective in RL is to learn a policy $\pi : \mathcal{S} \mapsto \Delta(\mathcal{A})$ that maps states to a probability distribution over actions in order to maximize the cumulative discounted reward starting from any initial state $s$. This objective can be expressed through the value function:

$$V^\pi(s) = \mathbb{E}_{a_t \sim \pi(s_t)} \left[ \sum_{t=0}^{T} \gamma^t r(s_t, a_t) \mid s_0 = s \right]. \tag{1}$$

The action-value function, or Q-function, further refines this by evaluating the expected return following a specific action $a$ in state $s$, under policy $\pi$:

$$Q^\pi(s, a) = \mathbb{E}_{a_t \sim \pi(s_t)} \left[ \sum_{t=0}^{T} \gamma^t r(s_t, a_t) \mid s_0 = s, a_0 = a \right], \tag{2}$$

and deep RL algorithms often use function approximators, such as neural networks, to estimate the Q-function and conduct policy learning. We denote the parameterized policy as $\pi_\theta$, where $\theta$ represents the parameters of the network.

Despite the effectiveness of RL algorithms such as proximal policy optimization (Schulman et al., 2017) and soft actor-critic (Haarnoja et al., 2018), these methods often suffer from high sample complexity (Wang et al., 2022), necessitating numerous interactions with the environment for convergence. In real-world applications, this requirement can be mitigated through engineering optimizations that reduce the cost of environmental interactions, allowing the training of competent agents at lower overall costs (Liang et al., 2018). This is particularly viable in Game AI contexts, where interactions with the environment are mediated by a game engine that enables rapid, large-scale parallel sampling (Vinyals et al., 2017; Guss et al., 2019). Such a setup facilitates the development of well-trained online policies at reduced costs, achieving high returns under specified reward functions and enhancing practical applications in real-world gaming environments (Jayaramireddy et al., 2022).

## 3.2 Representing and Learning from Human Preferences

It is crucial for effective integration to align agents with human users by accurately and efficiently identifying and representing human preferences. A traditional approach to understanding these preferences involves using inverse reinforcement learning (IRL) to infer reward functions that explain observed human behaviors (Ab Azar et al., 2020). However, IRL faces challenges such as high sample complexity, often requiring extensive demonstration data to accurately infer reward functions (Adams et al., 2022; Liu et al., 2024). Additionally, IRL can suffer from ambiguity, as multiple reward functions can explain the same observed behavior, limiting its ability to reflect human preferences accurately (Arora & Doshi, 2021).

Recent advancements in LLM alignment offer a promising alternative for recognizing human preferences. By employing a transformer-structured reward model, preferences can be learned directly from data through pairwise comparisons of human choices (Ouyang et al., 2022; Wang et al., 2023). The objective function for training such a reward model can be formulated as:

$$L_\phi = -\mathbb{E}_{(x, y_w, y_l) \sim \mathcal{D}} \left[ \log(\sigma(r_\phi(x, y_w) - r_\phi(x, y_l))) \right] \tag{3}$$

where $r_\phi$ represents the reward model parameterized by $\phi$ and $(x, y_w, y_l)$ denotes a sample triplet consisting of an input prompt $x$, a preferred (winning) completion $y_w$, and a less preferred (losing) completion $y_l$. The dataset $\mathcal{D}$ comprises such comparisons. This training objective leverages a contrastive learning format, demonstrating effectiveness in encoding human preferences. The transformer-based architecture of the reward model further enhances its ability to scale, efficiently handling long sequences and complex structures (Zhong et al., 2024; Wang et al., 2024a).

## 4 Online-to-Offline RL for Agent Alignment

In this section, we describe our approach to aligning online-trained reinforcement learning agents with human preferences using offline human data. The proposed approach utilizes two primary

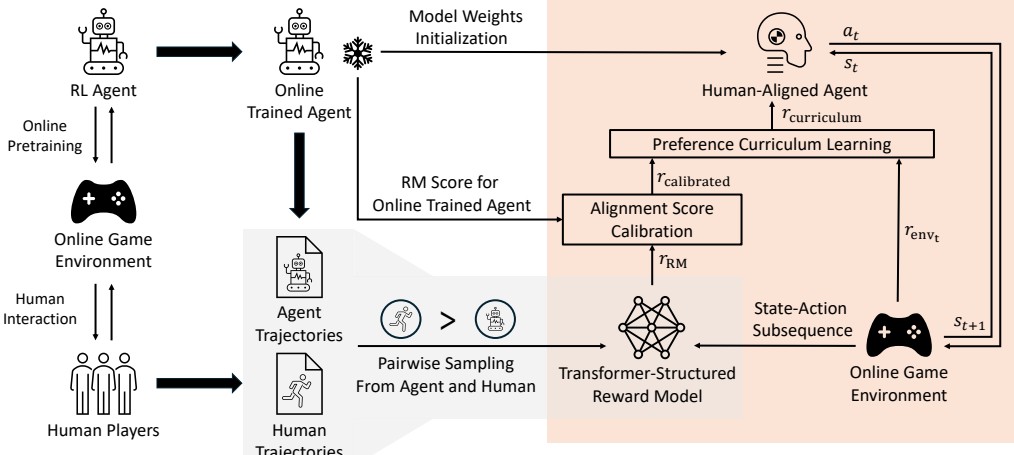

Figure 2: The framework of ALIGN-GAP for aligning RL game agent with human preferences. The gray section represents the training of the reward model, while the orange section represents the calibrated curriculum preference learning component.

strategies: First, we extract human preferences embedded in offline data by employing a transformer-based reward model (Section 4.1). Following the development of the reward model, we identify several challenges associated with aligning agents to these extracted preferences. To address these challenges, we introduce a calibrated preference curriculum learning approach. This strategy focuses on identifying behaviors that align with human preferences but are not yet adopted by the agent and transitions the agent's focus from maximizing environmental rewards to aligning with human preferences (Section 4.2). The framework and process of our method are illustrated in Figure 2.

## 4.1 Preference Extraction from Online Agent and Offline Human Data

**Extracting preferences with reward model.** To align the behaviors and styles of an online-trained agent with human preferences, the initial step involves extracting these preferences from offline human data, particularly in comparison to the behaviors and styles of the online agent. In the realm of LLM alignment, preference extraction can be achieved by training a reward model, typically structured as a transformer with a linear head for predicting reward values (Achiam et al., 2023). This architecture effectively captures contextual information and provides robust preference predictions (Ouyang et al., 2022; Wang et al., 2024a). In reinforcement learning, human behavioral preferences can be represented through sequential decision-making across various states. Similarly, a transformer structure can help capture contextual information within trajectories. As discussed previously, the reward model in LLM alignment is trained using triplet data $(x, y_w, y_l)$, where $x$ represents a prompt, and $y_w$ and $y_l$ denote the winning and losing responses, respectively. However, the scenario differs in reinforcement learning environments: instead of generating tokens based on a given prompt, the dynamics involve continuous updates of states and actions as the agent interacts with the environment (Li et al., 2023). To adapt this methodology, we train the reward model $R_\phi$ for agent alignment using a similar loss function while constructing the input to consist of state-action sequences from offline human data and trajectories sampled from online agent interactions. Specifically, the loss function for training the reward model is defined as:

$$L_\phi = -\log \mathbb{P}(\tau_{\text{human}} > \tau_{\text{agent}}) = -\mathbb{E}_{(\tau_{\text{human}}, \tau_{\text{agent}}) \sim \mathcal{D}} \left[\log(\sigma(R_\phi(\tau_{\text{human}}) - R_\phi(\tau_{\text{agent}})))\right] \quad (4)$$

where $\tau_{\text{human}} = \{(s_{\text{human}}, a_{\text{human}})\}_1^T$ represents a state-action sub-trajectory sampled from offline human data, and $\tau_{\text{agent}} = \{(s_{\text{agent}}, a_{\text{agent}})\}_1^T$ is the sub-trajectory of behaviors from the agent, which interacts with the environment using an epsilon-greedy strategy. Notably, unlike the environment reward function, the learned model $R_\phi$ does not take one single state-action pair as input but instead uses behavior sub-trajectories. This training approach is commonly referred to as preference-based reinforcement learning (PbRL) (Guan et al., 2022; Dong et al., 2023), building upon prior models but differing in dataset construction strategies and initial network configurations.

## 4.2 CURRICULUM PREFERENCE LEARNING WITH CALIBRATED RM SCORE

After obtaining the reward model, a straightforward approach might be to use it as the new reward function for fine-tuning the existing online well-trained agent. However, we find experimentally that this direct application often leads to sub-optimal performance, resulting in ineffective alignment with the preferences encoded in the reward model and undesirable unlearning of previously acquired behaviors (Nakamoto et al., 2024). We believe this outcome is likely due to the significant shift in objectives for the agent, from maximizing environmental returns to aligning with human preferences. This motivates us to develop a method for smoother alignment of online-trained agents with human preferences. From our experiments, we identify two main challenges:

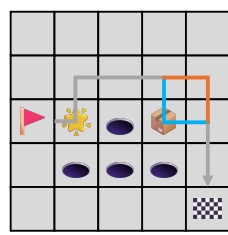

Figure 3: Example of trajectories from the online-trained agent and human achievement hunter.

**1) Identification of what to learn:** The reward model provides scores that reflect human preferences, but a crucial aspect is determining "what is preferred by humans but not yet learned by the agent." Given that the agent has undergone extensive online training, it already has an understanding of environmental dynamics. Our goal is to guide the agent towards decisions that align more closely with human preferences. For instance, in Figure 3, while the reward model may assign high rewards to human-preferred behaviors (indicated by the gray and blue trajectories), we aim for the preference signal used for alignment to prioritize actions that the agent has not yet learned to adopt (shown by the blue trajectory) over those it has already mastered (the gray trajectory).

**2) Mitigating sharp reward switching:** Directly applying the trained reward model as a new reward function for reinforcement learning fine-tuning often leads to a significant initial drop in performance, indicative of agent unlearning (Nakamoto et al., 2024). This can occur due to the sharp transition from a reward function that maximizes environmental returns to one that aligns with human preferences, creating a substantial gap between the agent's learned Q-function and the optimal Q-function for preference alignment (Gleave et al., 2020).

**RM score calibration with initial behavior baseline.** To address the first issue, we calibrate the reward signals used for preference learning to focus on aspects preferred by humans but not yet aligned by the agent. The reward for alignment should adapt to both human behavior and the style of the online-trained agent. We begin by evaluating the behavior of the online-trained policy using the reward model and using this assessment as a baseline for calibrating the preference signal:

$$R_{\text{calibrated}}(\tau) = R_\phi(\tau) - R_\phi(\tau'), \tag{5}$$

where $\tau \sim \{(s_t, \pi_\theta(s_t))\}_{t=1}^T$ represents the state-action trajectory sampled from the current policy $\pi_\theta$, and $\tau'$ is derived by replacing the actions in $\tau$ with those taken under the same states by the initial policy $\pi_0$ during the alignment phase. This calibrated reward tends to assign higher rewards to human-preferred behaviors that the agent has not yet learned, while the rewards for behaviors already aligned with human preferences approach zero. Since the alignment phase's policy initially matches $\pi_0$, under calibrated rewards, positive rewards imply that the agent's alignment is at least not worse than that of the online-trained policy.

**Reward curriculum for smoothed preference learning.** To address the second issue, we observed that even with calibrated rewards, the distribution of the reward function during the agent's online training can differ significantly, potentially leading to agent unlearning. In reinforcement learning, transitioning between different reward functions often involves transfer learning strategies to ensure smooth adaptation. It is important to note that transfer reinforcement learning typically addresses similar reward functions, such as perturbations of a single reward function (Zhu et al., 2023), and requires careful design to prevent unlearning when faced with significantly different reward functions (Gleave et al., 2020).

Considering that the agent is already well-trained to maximize environmental returns, preference learning presents a novel and potentially more challenging task. Therefore, we can design reward curriculum that gradually transitions the task represented by the reward function from maximizing

environmental returns to learning preferences:

$$R_{\text{curriculum}} = (1 - \alpha(t)) \cdot r_{\text{env}} + \alpha(t) \cdot R_{\text{calibrated}}, \tag{6}$$

where $r_{\text{env}}$ is the immediate reward feedback from the environment, $R_{\text{calibrated}}$ is the calibrated reward defined above, and $\alpha(t) : t \mapsto [0, 1]$ balances the ratio of environmental rewards and calibrated RM rewards, which can usually grow from 0 to 1. For instance, a linear transition can be defined as $\alpha(t) = \frac{\text{current\_step}}{\text{total\_step}}$. This reward curriculum facilitates a gradual transition from the familiar task of environmental reward maximization to the novel task of human preference alignment, offering a general solution for the learning process. We provide an analysis from theoretical perspective in Appendix A.2 for the effect of linear preference curriculum. In Section 5.4, we explore the effectiveness of different strategies of adjusting $\alpha(t)$ for reward curriculum design.

## 5 EXPERIMENTS

In this section, we aim to address the following key questions through experiments: (i) How effectively can our proposed method align online agents with human preferences represented with oracle rewards (Section 5.2)? (ii) How effectively can our proposed method align with human preferences from unsupervised classified human data (Section 5.3)? (iii) How do various components of our method, such as reward model calibration and preference curriculum, impact the overall alignment performance (Section 5.4)?

### 5.1 EXPERIMENTAL SETUP

**Environments and human preferences.** In our experiments, we evaluate various environments and constructions of human preferences. Specifically, we conduct experiments on the D4RL locomotion tasks, including HalfCheetah, Walker2D, and Hopper environments (Fu et al., 2020). These environments are control-oriented and resemble action-based gaming scenarios. For each environment, we train online agents using the reward functions provided by the respective environments.

To construct different human preferences, we design reward weights based on the environment rewards in the locomotion tasks. The default preference weights are set to $[1, 1, 1]$, reflecting three dimensions: maintaining health, moving forward, and controlling costs (Fu et al., 2020). We explore four combinations of preference weights: A: $[1, 0, 1]$, B:$[1, 1, 0]$, C: $[1, 10, 1]$, and D:$[1, 1, 10]$. These combinations yield distinct reward oracles that are solely used to train human proxies and evaluate the aligned agents. After training the human proxies, we collect a small amount of human data (only 10 episodes) using the proxies for subsequent agent alignment. In Appendix A.1, we provide a visual analysis of the different styles exhibited by various proxies across environments.

Additionally, we extend our experiments to Atari Pac-Man and Space Invaders games (Brockman, 2016). To assess the performance of our proposed methods against various constructions of human preferences, we utilize Google's Atari Replay Datasets (Agarwal et al., 2020) and conduct unsupervised classification on the dataset using UMAP (McInnes et al., 2018). In each environment, we obtain three distinct preferences of human data, retaining only 10 episodes for each preference. The online-trained agent is then aligned with the preferences encoded in the human data.

**Comparisons, baselines, and evaluation protocol.** For the D4RL locomotion tasks, we employ Soft Actor-Critic (SAC) (Haarnoja et al., 2018) to train our online well-trained agents. In the Atari games, we utilize Deep Q-Networks (DQN) (Roderick et al., 2017) for online agent training. We compare our approach against a fine-tuning strategy with SAC (termed Finetune w/ SAC) or DQN (termed Finetune w/ DQN) that leverages offline data alongside the online training process. Specifically, we adopt symmetric sampling, where we sample 50% from the replay buffer containing offline data and 50% from online data. This method has been shown to yield better results compared to using only offline data (Ball et al., 2023).

Additionally, we evaluate a baseline that trains the agent using both online environments and offline human data, mimicking DDPGfD (Vecerik et al., 2017) but employing SAC (termed SACfD) or DQN (termed DQNfD) instead of DDPG, to match the algorithm used during the agent's online learning phase. Further details and hyperparameter settings can be found in Appendix A.4. We also assess the effectiveness of Behavior Cloning (BC) (Torabi et al., 2018) as a baseline, evaluating how

Table 1: Returns for online trained agent and human performance, and **differences in return to human** for different alignment approaches. Results are averaged over 5 random seeds.

| Task-Preference | Online Trained | Human Performance | Finetune w/ SAC | Finetune w/ BC | SACfD | BC | Finetune w/ RM | ALIGN-GAP (Ours) |
|---|---|---|---|---|---|---|---|---|
| Halfcheetah-A | | 2.2±0.01 | 101.77±1.2 | 20.61±24.04 | 52.08±0.57 | -4.07±0.87 | -2.51±0.03 | -0.86±0.33 |
| Halfcheetah-B | 103.05±0.67 | 125.32±0.76 | -2.68±1.78 | -31.25±9.81 | -113.9±13.14 | -125.99±0.65 | -14.59±36.82 | -0.44±1.97 |
| Halfcheetah-C | | 105.27±3.3 | 3.0±0.44 | -8.7±12.33 | -100.23±3.12 | -105.72±1.42 | -105.39±0.03 | -0.91±0.86 |
| Halfcheetah-D | | 114.74±0.45 | -39.62±8.56 | -13.5±5.94 | -109.95±2.58 | -116.81±0.37 | -115.26±0.25 | -7.3±0.29 |
| Walker2d-A | | 20.72±2.06 | 89.65±1.0 | -2.95±5.57 | 67.91±0.5 | -19.45±1.41 | 2.65±0.03 | 2.22±3.1 |
| Walker2d-B | 112.23±0.69 | 122.11±0.42 | -1.54±1.38 | -50.88±44.76 | -9.59±0.48 | -119.78±2.72 | -100.4±0.03 | -2.87±26.06 |
| Walker2d-C | | 116.53±0.74 | 1.57±0.71 | -81.21±44.32 | -18.94±1.46 | -114.69±1.54 | -74.54±11.61 | -1.43±1.78 |
| Walker2d-D | | 97.35±26.53 | 16.1±0.58 | -16.42±39.58 | 12.55±0.65 | -86.08±5.49 | -75.56±0.19 | -3.04±27.97 |
| Hopper-A | | 30.74±0.01 | 63.82±30.29 | 0.01±0.01 | 79.25±0.19 | -23.3±3.49 | -30.09±0.01 | 6.46±1.56 |
| Hopper-B | 88.15±38.98 | 93.88±28.17 | 6.7±34.8 | -5.13±22.83 | -33.33±22.63 | -75.74±19.89 | -93.15±0.0 | -3.33±0.02 |
| Hopper-C | | 31.15±4.44 | 0.61±4.99 | -0.05±1.47 | 1.24±2.7 | -7.71±11.42 | -23.27±0.63 | 0.57±2.64 |
| Hopper-D | | 110.07±0.35 | -18.57±29.14 | -9.22±21.58 | -44.88±30.56 | -105.18±5.57 | -78.14±0.02 | -8.77±0.49 |

Table 2: Ranking the alignment degree of different alignment approaches by their similarity to human under oracle rewards (preferences). Results are averaged over 5 random seeds.

| Task-Preference | Online Trained | Finetune w/ SAC | Finetune w/ BC | SACfD | BC | Finetune w/ RM | ALIGN-GAP (Ours) | ALIGN-GAP w/o Calibration | ALIGN-GAP w/o Curriculum |
|---|---|---|---|---|---|---|---|---|---|
| Halfcheetah-A | Rank 9 | Rank 8 | Rank 4 | Rank 5 | Rank 3 | Rank 2 | Rank 1 | Rank 6 | Rank 7 |
| Halfcheetah-B | Rank 9 | Rank 3 | Rank 8 | Rank 7 | Rank 6 | Rank 4 | Rank 1 | Rank 2 | Rank 5 |
| Halfcheetah-C | Rank 9 | Rank 3 | Rank 7 | Rank 5 | Rank 6 | Rank 8 | Rank 1 | Rank 2 | Rank 4 |
| Halfcheetah-D | Rank 9 | Rank 3 | Rank 5 | Rank 6 | Rank 8 | Rank 7 | Rank 1 | Rank 2 | Rank 4 |
| Walker2d-A | Rank 9 | Rank 8 | Rank 5 | Rank 6 | Rank 7 | Rank 4 | Rank 2 | Rank 1 | Rank 3 |
| Walker2d-B | Rank 9 | Rank 4 | Rank 6 | Rank 3 | Rank 7 | Rank 8 | Rank 1 | Rank 2 | Rank 5 |
| Walker2d-C | Rank 9 | Rank 3 | Rank 8 | Rank 2 | Rank 6 | Rank 5 | Rank 1 | Rank 4 | Rank 7 |
| Walker2d-D | Rank 9 | Rank 4 | Rank 2 | Rank 3 | Rank 5 | Rank 6 | Rank 1 | Rank 8 | Rank 7 |
| Hopper-A | Rank 9 | Rank 7 | Rank 5 | Rank 8 | Rank 2 | Rank 6 | Rank 3 | Rank 1 | Rank 4 |
| Hopper-B | Rank 9 | Rank 2 | Rank 5 | Rank 4 | Rank 6 | Rank 8 | Rank 1 | Rank 3 | Rank 7 |
| Hopper-C | Rank 9 | Rank 3 | Rank 5 | Rank 4 | Rank 6 | Rank 7 | Rank 1 | Rank 2 | Rank 8 |
| Hopper-D | Rank 9 | Rank 3 | Rank 6 | Rank 5 | Rank 7 | Rank 8 | Rank 1 | Rank 2 | Rank 4 |

directly applying cloning on limited human data aligns with human preferences (termed Finetune w/ BC for behavior cloning with policy initialized from online agent, and BC for behavior cloning from scratch). Moreover, we investigate the performance of agents aligned by directly using the trained RM as the reward function (termed Finetune w/ RM), comparing with our proposed approach that incorporates reward calibration and reward curriculum.

During evaluation, we first assess the effectiveness of agent alignment by examining the performance of both the online-trained agents and human proxies within the context of environmental rewards. Specifically, we evaluate how well these agents perform in the environment by comparing their returns against those achieved by human players. For each alignment method, we quantify its success by measuring the performance discrepancy between the agents aligned using that method and the human proxies that serve as benchmarks for human-like behavior. This analysis enables us to evaluate the extent to which the performance of the aligned agents aligns with human preferences, as reflected in the differences in their environmental rewards.

Given that human preferences are inherently subjective and relative, we utilize the reward oracles from training the human proxies as representatives of human preferences for locomotion tasks. We assess the relative performance of various alignment approaches by ranking their similarity to the human proxy under the oracle rewards. Additionally, we visualize the trajectory styles exhibited by different methods in both locomotion and Atari tasks for subjective evaluation and comparison.

## 5.2 Aligning Agents with Preference Data from Human Proxy

To assess the alignment effectiveness of the online well-trained agents with human preferences, we first focus on the performance gaps between the agents and human proxies within locomotion tasks. In Table 1, we present the returns of both the online-trained agents (labeled "Online Trained") and the human proxies (labeled "Human Performance"). Additionally, for the baseline alignment approaches and ALIGN-GAP, the table displays the differences in returns against human under environmental rewards. Smaller differences in this metric indicate that the agents perform in a manner that can be more consistent with human behavior in the game environment. Our proposed method demonstrates smaller gaps across various environments and preference types compared to the baselines, highlighting its alignment capability.

To evaluate the effectiveness of our alignment from the human perspective, we use the reward oracle that trained the human proxy to assess various agents. Table 2 presents the rankings based on the differences in performance between the methods and the human proxy under oracle rewards. A higher ranking indicates that an agent's behavior aligns more closely with the human preferences implied by the oracle rewards. ALIGN-GAP significantly outperforms other methods, achieving top rankings across various environments and preference settings. To further validate the reliability of these rankings, we compare agent trajectories generated by different alignment methods against those of the human proxy. This qualitative analysis is complemented by a quantitative evaluation using Spearman's rank correlation coefficient (Sedgwick, 2014). Spearman's coefficient measures the degree of rank-order correlation between two variables, with a value of $0.87$ observed between rankings derived from oracle rewards and human evaluations. This high correlation substantiates the alignment results reported in Table 2. An example of different human proxy preferences and the styles learned by ALIGN-GAP is shown in Figure 9 of Appendix A.1.

## 5.3 ALIGNING AGENTS WITH UNSUPERVISED CLASSIFIED HUMAN DATA

In the Atari tasks, we focus on aligning the online well-trained agents with unsupervised classified human data, processed through UMAP to represent the diverse preferences inherent within this dataset. Evaluations based on performance gaps in environmental rewards similar to those outlined in Table 1 are listed in Appendix A.3.

To further assess the manifestation of human preferences in clustered data and the alignment efficacy of ALIGN-GAP, Figure 4a presents the UMAP clustered results for Space Invaders. Besides, we conduct qualitative evaluations by visualizing both the categorized human preferences and the behaviors learned by ALIGN-GAP, with Space Invaders serving as an example (as depicted in Figure 5). Observations reveal that while the online-trained agent typically adopts a strategy focused on eliminating the majority of invaders, human players exhibit varied approaches. For example, some may focus on attacking specific columns or strategic areas of the screen. Style 1 players might prioritize attacking the second and fourth columns of invaders, indicative of a patterned approach to game-play, whereas Style 2 players may target invaders predominantly on the left side, perhaps reflecting a preference for handling imminent threats or a specific game-play strategy.

## 5.4 ANALYSIS ON THE COMPONENTS OF ALIGN-GAP

**Ablation Study** Table 2 shows the performance of our proposed method without RM score calibration or preference curriculum. The results clearly indicate that the alignment effectiveness is significantly diminished in the absence of the curriculum, demonstrating a substantial impact on performance. This highlights the critical role of both the calibrated preference curriculum and RM score calibration in ensuring robust and effective alignment of agents with human preferences.

Additionally, regarding the design of $\alpha(t)$ in the preference curriculum, we explore alternative adjustment methods beyond the linear approach. Specifically, we evaluate an exponential form of $\alpha(t) = 1 - \exp(-\lambda \cdot \frac{\text{current\_step}}{\text{total\_step}})$, where $\lambda = 5$ is a positive constant to ensure that $\alpha(t)$ follows the trend of increasing from 0 to approaching 1. The results for the HalfCheetah environment are presented in Table 3. The findings indicate that both linear and exponential adjustments yield similar performance, demonstrating the robustness of the preference curriculum method.

Table 3: Comparing different $\alpha(t)$ for preference curriculum learning on HalfCheetah environment. The evaluation metrics remain consistent with Table 1.

| Preference | Linear | Exponential |
|---|---|---|
| A | -0.86±0.33 | -0.95±0.48 |
| B | -0.44±1.97 | -0.93±0.76 |
| C | -0.91±0.86 | -0.80±1.13 |
| D | -7.3±0.29 | -7.9±0.14 |

**What is learned from the reward model?** After training the reward model, evaluating its effectiveness in reflecting human preferences embedded in the data is essential. To do this, we assess how well the reward model's predictions align with human preferences using trajectories from an unseen test set. Specifically, we analyze whether the model accurately assigns preferences to these trajectories consistent with ground truth expectations. The results, illustrated in Figure 4b, demonstrate that the

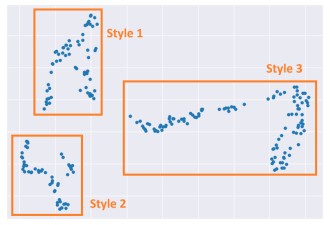
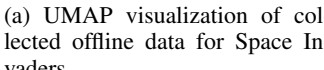

(a) UMAP visualization of collected offline data for Space Invaders.

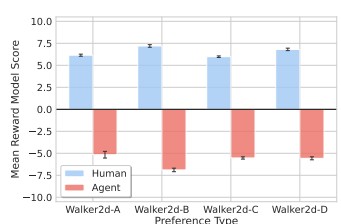
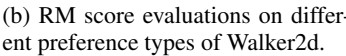

(b) RM score evaluations on different preference types of Walker2d.

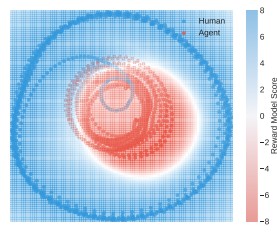

(c) The reward distribution learned by the reward model.

Figure 4: Analysis of reward distribution for RM and offline human data distribution.

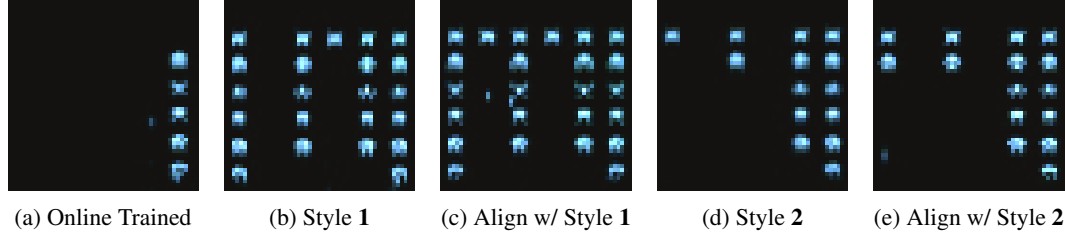

(a) Online Trained     (b) Style **1**     (c) Align w/ Style **1**     (d) Style **2**     (e) Align w/ Style **2**

Figure 5: State of remaining invaders at the end of a Space Invaders game for an online-trained agent, two styles of offline behavior data, and the corresponding ALIGN-GAP aligned agents. All episodes used the same random seed to initialize the episode and game engine.

model successfully prefers human trajectories over those of the online-trained agent, indicating that it effectively captures the nuances of human-like behavior.

Moreover, to visually explore the stylistic differences between human data and online-trained agent trajectories, we utilize PCA (Maćkiewicz & Ratajczak, 1993) for dimension reduction on both datasets. We then create a heat map of the reward model's scores by querying the model with data points reconstructed using the inverse mapping from PCA. Presented in Figure 4c, this visualization displays the reward model's distribution of scores across the reduced dimensional space. The PCA highlights clear distinctions between the human data (depicted in blue points) and the online-trained agent data (shown in red points). Notably, areas of proximity in this space mainly reflect similarities between agent and human behaviors during the initial stages of episodes. The heat map further reveals that the reward model assigns higher rewards to points closely resembling human trajectory data and lower rewards to those more aligned with the behaviors exhibited by the online-trained agent. This pattern emphasizes the reward model's ability to effectively discriminate and prioritize human-like behaviors, confirming its capacity to guide agents towards more human-aligned actions and decision-making styles.

## 6   CONCLUSION, LIMITATIONS AND FUTURE WORK

In this paper, we formulate the problem of *online-to-offline* RL for agent alignment and introduce ALIGN-GAP to align agents with human preferences in game environments. While ALIGN-GAP demonstrates strong alignment capabilities, we believe that achieving more fine-grained alignment is possible by carefully designing the calibration for specific environments. Our experiments focus on gaming due to the availability of trained agents and preference data, and future work may extend to broader domains such as autonomous driving, healthcare, and multi-agent systems, where aligning AI with human values remains crucial.

### ACKNOWLEDGMENTS

The corresponding author Shuai Li is sponsored by CCF-Tencent Open Research Fund.

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

# A APPENDIX

## A.1 STYLES OF HUMAN PREFERENCES IN HALFCHEETAH, WALKER2D AND HOPPER

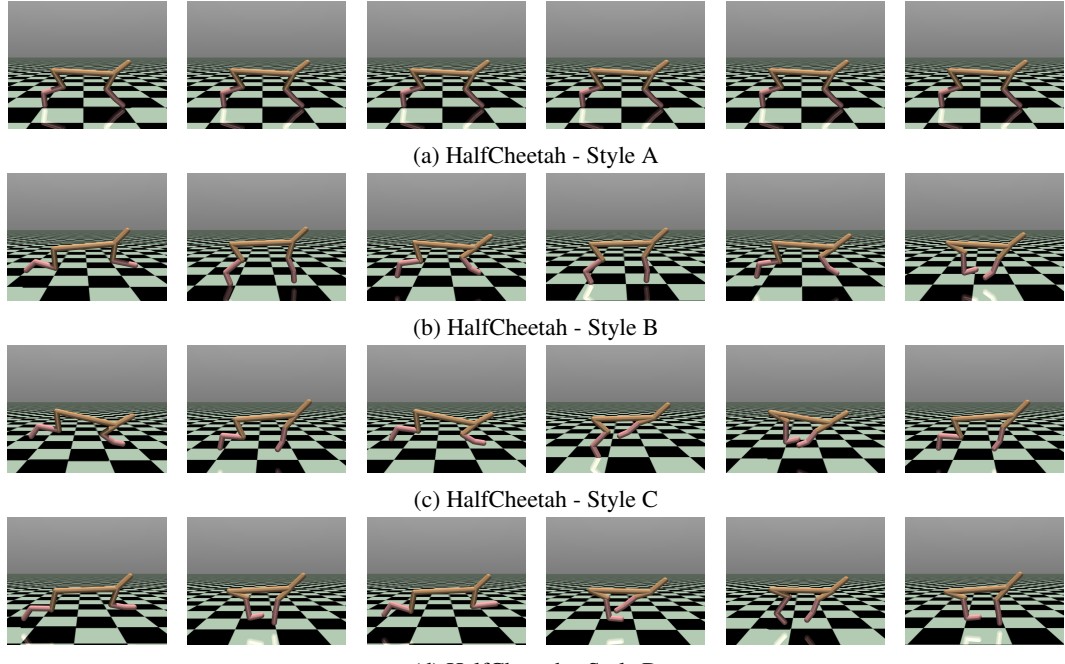

(a) HalfCheetah - Style A

(b) HalfCheetah - Style B

(c) HalfCheetah - Style C

(d) HalfCheetah - Style D

Figure 6: HalfCheetah Styles.

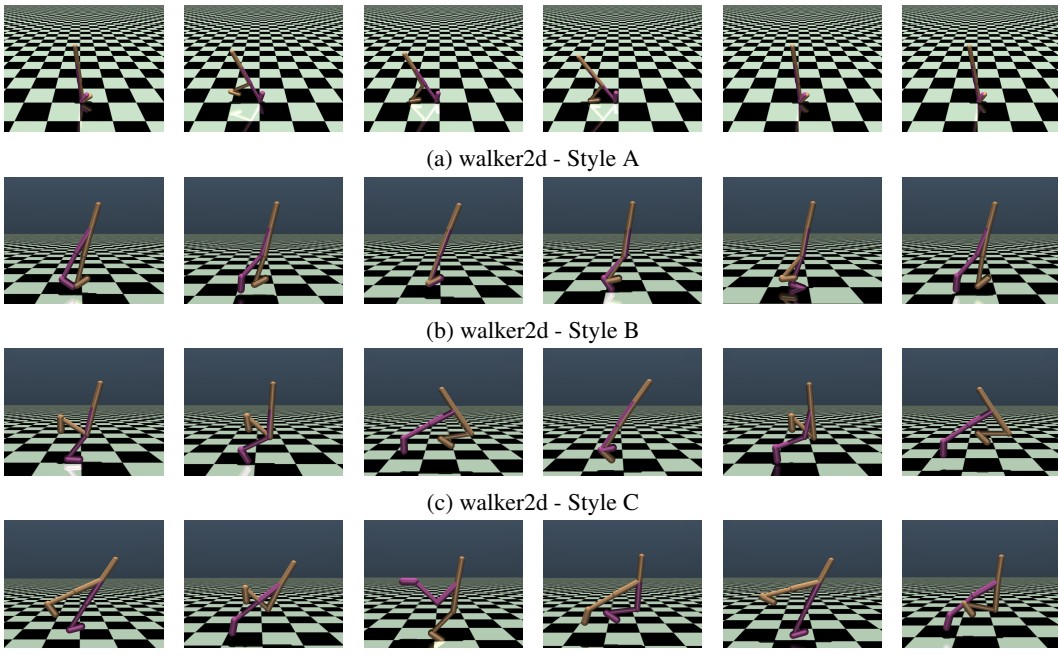

(a) walker2d - Style A

(b) walker2d - Style B

(c) walker2d - Style C

(d) walker2d - Style D

Figure 7: Walker2d Styles.

In Figure 6, 7, and 8, we showcase examples of different styles exhibited by human proxies trained under various reward oracles. In different environments, human styles may manifest in movement

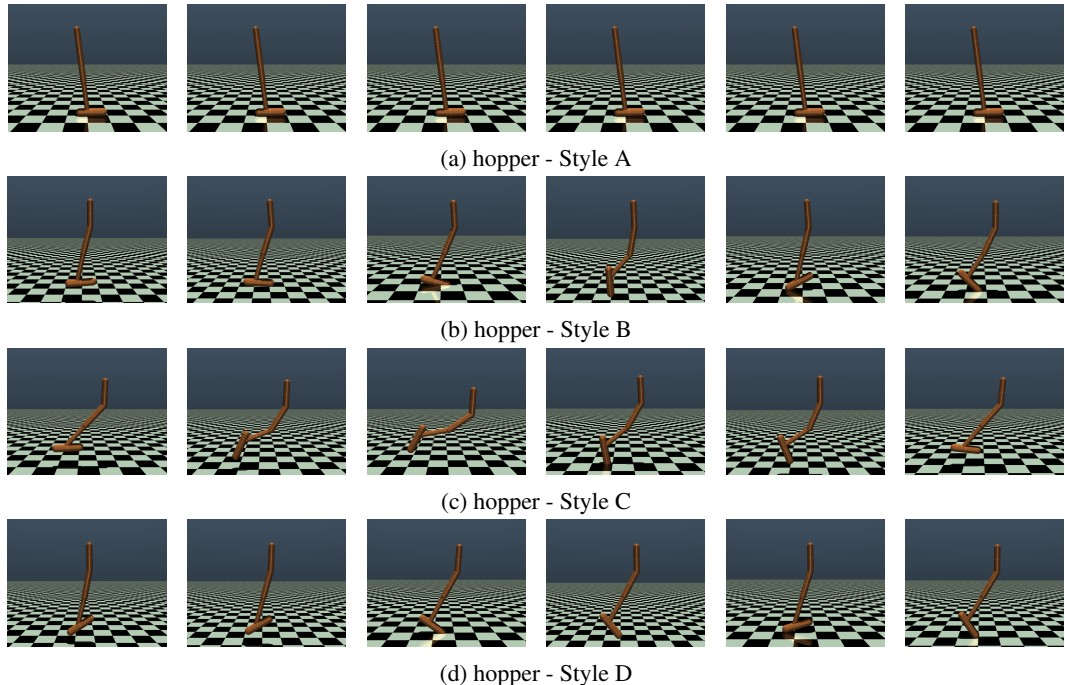

(a) hopper - Style A

(b) hopper - Style B

(c) hopper - Style C

(d) hopper - Style D

Figure 8: Hopper Styles.

speed, special postures during movement, evasive action styles in special situations, and so on. Using Walker2D and Hopper environment as an example, we demonstrate the distinctions between different human styles in terms of specific stylistic features, as well as ALIGN-GAP's learning effectiveness for these styles, and an example is shown in Figure 9.

In Figure 9, we showcase the varied styles of human preferences in the Walker2D and Hopper environments and what is learned with offline data of those preferences by ALIGN-GAP. The Walker2D examples illustrate distinct running styles: one proxy exhibits a long-stride running pattern, characterized by extended leg movements to cover more ground per stride, while another demonstrates a forward-leaning running style, emphasizing speed and forward momentum with an aggressive stance. In the Hopper environment, contrasting jumping styles are evident; one style is marked by large-amplitude jumps that prioritize vertical movement and height, whereas the other features small-step jumping, focusing on stability and control with shorter, more frequent leaps. We also conduct visualizations on the styles learned by the agent with ALIGN-GAP, showing similar styles to that of human. It can be seen that by using ALIGN-GAP for agent alignment, the agent could recognize the preference of human and learn from human styles in offline data to improve its behaviors.

## A.2 THEORETICAL JUSTIFICATION FOR REWARD CURRICULUM OF ALIGN-GAP

In this section, we provide an analysis from theoretical perspective on the effect of linear $\alpha(t)$ for preference curriculum, focusing on its effect for bridging two reward functions, the reward function $R_{\text{env}}$ from the environment and the reward function $R_{\text{human}}$ for measuring the alignment degree of the agent. The analysis is first motivated from the reward function measure EPIC (Gleave et al., 2020), where we can measure the "distance" between two reward functions.

**Definition A.1** (Canonically Shaped Reward). Let $R : \mathcal{S} \times \mathcal{A} \times \mathcal{S} \to \mathbb{R}$ be a reward function. Given distributions $\mathcal{D}_\mathcal{S} \in \Delta(\mathcal{S})$ and $\mathcal{D}_\mathcal{A} \in \Delta(\mathcal{A})$ over states and actions, let $S$ and $S'$ be random variables independently sampled from $\mathcal{D}_\mathcal{S}$ and $A$ sampled from $\mathcal{D}_\mathcal{A}$. Define the canonically shaped $R$ to be:

$$C_{\mathcal{D}_\mathcal{S}, \mathcal{D}_\mathcal{A}}(R)(s, a, s') = R(s, a, s') + \mathbb{E}[\gamma R(s', A, S') - R(s, A, S') - \gamma R(S, A, S')]. \quad (7)$$

Based on the canonically shaped reward, we can define the EPIC pseudometric that measures the distance between two reward functions (Gleave et al., 2020).

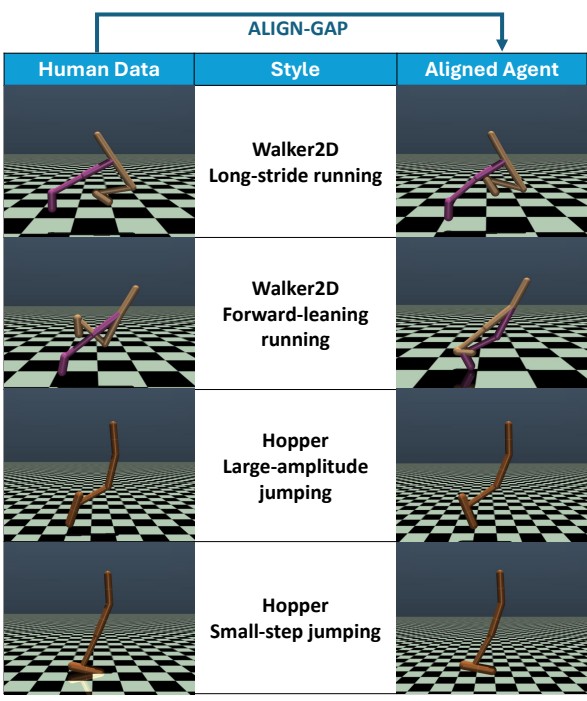

Figure 9: Different human styles in Walker2d and Hopper environment and the aligned agent by ALIGN-GAP. All episodes used the same random seed to initialise the episode and game engine.

**Definition A.2** (Equivalent-Policy Invariant Comparison (EPIC) pseudometric). Let $\mathcal{D}$ be some coverage distribution over transitions $s \xrightarrow{a} s'$. Let $S, A, S'$ be random variables jointly sampled from $\mathcal{D}$. Let $\mathcal{D}_S$ and $\mathcal{D}_A$ be some distributions over states $\mathcal{S}$ and $\mathcal{A}$ respectively. The Equivalent-Policy Invariant Comparison (EPIC) distance between reward functions $R_A$ and $R_B$ is:

$$D_{\text{EPIC}}(R_A, R_B) = D_\rho\left(C_{\mathcal{D}_S, \mathcal{D}_A}(R_A)(S, A, S'), C_{\mathcal{D}_S, \mathcal{D}_A}(R_B)(S, A, S')\right). \tag{8}$$

Previous work shows that the Equivalent-Policy Invariant Comparison distance is a pseudometric, and that let $R_A, R_B : \mathcal{S} \times \mathcal{A} \times \mathcal{S} \to \mathbb{R}$ be reward functions, then $0 \le D_{\text{EPIC}}(R_A, R_B) \le 1$. Now we will prove the main result for showing the reduction in distance between the environment rewards and the alignment target, by introducing a linear reward curriculum between them.

**Theorem A.3.** *Denote the reward function for the online game environment as $R_{\text{env}}$ and the reward function implied and learned from the reward model and calibrated in ALIGN-GAP as $R_{\text{human}}$. Assuming the reward functions are of unit variance (which can be realized in practice by normalization), then the EPIC distance for the constructed curriculum reward $R_{\text{align}}$ satisfies*

$$\mathbb{E}\left[\frac{D_{\text{EPIC}}(R_{\text{env}}, R_{\text{align}})}{D_{\text{EPIC}}(R_{\text{env}}, R_{\text{human}})}\right] \le \frac{2}{3}. \tag{9}$$

*Proof.* Considering the curriculum reward learning with parameter $\alpha$ as

$$R_{\text{align}} = \alpha R_{\text{env}} + (1 - \alpha) R_{\text{human}}, \tag{10}$$

where $\alpha \in [0, 1]$ is the curriculum hyper-parameter. Denoting the canonically shaped reward for environment rewards and human rewards as $C_{\text{env}}$ and $C_{\text{human}}$, respectively, since the definition of canonically shaped reward maintains linearity, we have the canonically shaped reward for $R_{\text{align}}$ (denoted as $C_{\text{align}}$) as also a linear combination of $C_{\text{env}}$ and $C_{\text{human}}$. By definition we have

$$D_\rho(C_{\text{env}}, C_{\text{human}}) = \frac{\sqrt{1 - \rho(C_{\text{env}}, C_{\text{human}})}}{\sqrt{2}}, \quad D_\rho(C_{\text{env}}, C_{\text{align}}) = \frac{\sqrt{1 - \rho(C_{\text{env}}, C_{\text{align}})}}{\sqrt{2}}. \tag{11}$$

Now we will first analyze the relations between $\rho(C_{\text{env}}, C_{\text{human}})$ and $\rho(C_{\text{env}}, C_{\text{align}})$:

$$
\begin{aligned}
\rho\left(C_{\text{env}}, C_{\text{align}}\right) &= \rho\left(C_{\text{env}}, \alpha C_{\text{env}} + (1-\alpha)C_{\text{human}}\right) \\
&= \frac{\operatorname{cov}\left(C_{\text{env}}, \alpha C_{\text{env}} + (1-\alpha)C_{\text{human}}\right)}{\sigma_{C_{\text{env}}} \sigma_{\alpha C_{\text{env}} + (1-\alpha)C_{\text{human}}}} \\
&= \frac{\alpha \sigma_{C_{\text{env}}}^2 + (1-\alpha)\operatorname{cov}\left(C_{\text{env}}, C_{\text{human}}\right)}{\sigma_{C_{\text{env}}} \sqrt{\alpha^2 \sigma_{C_{\text{env}}}^2 + (1-\alpha)^2 \sigma_{C_{\text{human}}}^2 + 2\alpha(1-\alpha)\operatorname{cov}\left(C_{\text{env}}, C_{\text{human}}\right)}} \\
&= \frac{\alpha \sigma_{C_{\text{env}}}^2 + (1-\alpha)\rho\left(C_{\text{env}}, C_{\text{human}}\right)\sigma_{C_{\text{env}}}\sigma_{C_{\text{human}}}}{\sigma_{C_{\text{env}}} \sqrt{\alpha^2 \sigma_{C_{\text{env}}}^2 + (1-\alpha)^2 \sigma_{C_{\text{human}}}^2 + 2\alpha(1-\alpha)\rho\left(C_{\text{env}}, C_{\text{human}}\right)\sigma_{C_{\text{env}}}\sigma_{C_{\text{human}}}}} \\
&= \frac{\alpha + (1-\alpha)\rho\left(C_{\text{env}}, C_{\text{human}}\right)}{\sqrt{\alpha^2 + (1-\alpha)^2 + 2\alpha(1-\alpha)\rho\left(C_{\text{env}}, C_{\text{human}}\right)}} \\
&\geq \alpha + (1-\alpha)\rho\left(C_{\text{env}}, C_{\text{human}}\right).
\end{aligned}
\tag{12}
$$

Since $\alpha, \rho\left(C_{\text{env}}, C_{\text{human}}\right) \in [0, 1]$, firstly we have $\rho\left(C_{\text{env}}, C_{\text{align}}\right) \geq \rho\left(C_{\text{env}}, C_{\text{human}}\right)$.

Furthermore, by the definition of the Pearson distance, it can be seen that

$$
\begin{aligned}
\frac{D_\rho\left(C_{\text{env}}, C_{\text{align}}\right)}{D_\rho\left(C_{\text{env}}, C_{\text{human}}\right)} &= \sqrt{\frac{1 - \rho\left(C_{\text{env}}, C_{\text{align}}\right)}{1 - \rho\left(C_{\text{env}}, C_{\text{human}}\right)}} \\
&\leq \sqrt{\frac{1 - \alpha - (1-\alpha)\rho\left(C_{\text{env}}, C_{\text{human}}\right)}{1 - \rho\left(C_{\text{env}}, C_{\text{human}}\right)}} \\
&= \sqrt{1 - \alpha}.
\end{aligned}
\tag{13}
$$

Therefore, by the non-negativity of the Pearson distance, we have

$$
\mathbb{E}\left[\frac{D_\rho\left(C_{\text{env}}, C_{\text{align}}\right)}{D_\rho\left(C_{\text{env}}, C_{\text{human}}\right)}\right] \leq \mathbb{E}\left[\sqrt{1 - \alpha}\right] = \int_0^1 \sqrt{1 - \alpha}\, d\alpha = \frac{2}{3}.
\tag{14}
$$

$\square$

*Remark* A.4. This result indicates that by introducing a linear reward curriculum, we can reduce the EPIC distance between different reward functions, thereby simplifying the learning difficulty of the agent's alignment process. For other forms of curricula, such as the exponential curriculum we also experimented with, similar proof processes can be employed to evaluate their potential advantages.

Moreover, given that there is typically a significant gap between the reward objectives for alignment with humans and the inherent reward objectives of the environment, we can derive the following inference under the assumption that these two are uncorrelated.

**Corollary A.5.** *If $R_{\text{env}}$ and $R_{\text{human}}$ are uncorrelated (i.e., $\operatorname{cov}\left(R_{\text{env}}, R_{\text{human}}\right) = 0$), then the curriculum reward learning in ALIGN-GAP mitigates the EPIC distance by nearly $50\%$.*

*Proof.* When the environment rewards and human preference rewards learned by the reward model are uncorrelated, we have

$$
D_{\text{EPIC}}\left(R_{\text{env}}, R_{\text{human}}\right) = \frac{\sqrt{1 - \rho(C_{\text{env}}, C_{\text{human}})}}{\sqrt{2}} = \frac{\sqrt{2}}{2},
\tag{15}
$$

which is the largest EPIC we could have for two different reward functions. In this case, from the deviation in Eq. (12), the relation between the aligned reward in ALIGN-GAP and the environment reward are now simpler:

$$
\rho\left(C_{\text{env}}, C_{\text{align}}\right) = \frac{\alpha}{\sqrt{\alpha^2 + (1-\alpha)^2}}, \quad \text{for } \alpha \in [0, 1].
\tag{16}
$$

Therefore, now we have

$$\mathbb{E}\left[D_{\text{EPIC}}\left(C_{\text{env}}, C_{\text{align}}\right)\right] = \frac{1}{\sqrt{2}} \int_0^1 \sqrt{1 - \frac{\alpha}{\sqrt{(1-\alpha)^2 + \alpha^2}}} \, d\alpha \approx 0.371 \qquad (17)$$

and ratio of the expected EPIC distance is

$$\mathbb{E}\left[\frac{D_{\text{EPIC}}\left(R_{\text{env}}, R_{\text{align}}\right)}{D_{\text{EPIC}}\left(R_{\text{env}}, R_{\text{human}}\right)}\right] = \frac{\mathbb{E}\left[D_{\text{EPIC}}\left(R_{\text{env}}, R_{\text{align}}\right)\right]}{D_{\text{EPIC}}\left(R_{\text{env}}, R_{\text{human}}\right)} \approx 52.5\%. \qquad (18)$$

$\square$

## A.3 ATARI RESULTS

In this section, we evaluate the effectiveness of different alignment methods in Atari domain (Pac-man, Space-invaders and Alien), and the result is shown in Table 4. Specifically, we analyze the performance of the online-trained agents and the performance of offline human data under various preferences. It is noteworthy that the offline data sampled within this dataset generally shows lower performance compared to the agents trained online, reflecting a gap between the two. For different alignment methods, we assess and record the difference in performance between the aligned agents and the human performance captured in the offline data. A smaller difference suggests that the aligned agent's behavior is more similar to human performance from a performance perspective. However, it is important to note that similarity in performance does not necessarily imply that the agent's style aligns with human preferences. In Section 5.2, we visually compare the styles of humans and the aligned agents to determine if the agents are well-aligned with human preferences.

Table 4: Returns for online trained agent and offline data performance, and **differences in return to offline data** for different alignment approaches. Results are averaged over 5 random seeds.

| Task-Preference | Online Trained | Offline Trejectory | Finetune w/ DQN | DQNfD | BC | Finetune w/ RM | ALIGN-GAP |
|---|---|---|---|---|---|---|---|
| Pac-Man-A |  | 148±12 | 1390±875 | 518±87 | 368±231 | 510±236 | 134±19 |
| Pac-Man-B | 2252±320 | 207±30 | 1636±808 | 546±246 | 144±56 | 462±197 | 132±25 |
| Pac-Man-C |  | 522±45 | 1398±826 | 90±20 | -190±22 | 116±37 | 54±18 |
| Spece-Invaders-A |  | 228±14 | 289±46 | 192±32 | 174±38 | 197±40 | 52±21 |
| Spece-Invaders-B | 656±108 | 431±38 | 188±85 | 152±68 | 180±62 | 60±38 | 96±27 |
| Spece-Invaders-C |  | 556±37 | 163±52 | -41±42 | -91±47 | -73±26 | 36±14 |
| Alien-A |  | 460±32 | 1731±92 | 468±61 | 328±76 | 178±28 | 56±19 |
| Alien-B | 3069±284 | 740±24 | 1890±74 | 565±87 | 236±20 | 147±50 | 37±30 |
| Alien-C |  | 982±76 | 1295±84 | 480±46 | 241±29 | 136±24 | 42±28 |

## A.4 HYPER-PARAMETERS

The hyper-parameters and values are summarized in Table 5.

Table 5: Hyper-parameters for agent alignment with ALIGN-GAP and baselines.

| Hyper-Parameters | Values |
|---|---|
| Online Pre-train Steps | 5e6 |
| Offline Alignment Steps | 1e6 |
| SAC Actor Learning Rate | 3e-4 |
| SAC Critic Learning Rate | 3e-4 |
| SAC Batch Size | 256 |
| Offline Trajectory Number | 10 |
| DQN Learning Rate | 5e-5 |
| DQN Target Update Interval | 10000 |
| DQN Batch Size | 32 |
| Reward Model Sequence Length | 64 |
| Reward Model Latent Dim | 256 |
| Reward Model Training Batch Size | 64 |
| Reward Model Learning Rate | 1e-4 |

## A.5 COMPARATIVE STUDY WITH RLPD-LIKE RE-TRAINING APPROACHES

To provide a comprehensive evaluation of ALIGN-GAP, we conducted a comparative study against RLPD-like approaches that align agents without leveraging pre-trained models (Ball et al., 2023). In this approach, the agent is trained from scratch using a mix of offline human preference data and environmental interactions, and the rewards come from the trained reward model to encode human preferences. This comparison is particularly relevant in understanding the trade-offs between training a new agent versus fine-tuning an existing well-trained agent for human alignment.

In the RLPD framework, offline data and online data are mixed according to a specified ratio during training. Following the recommendations in RLPD, we employed symmetric sampling, where offline and online data are sampled equally (50% each). To ensure a fair comparison, we evaluated RLPD under two settings:

1. **1e6 alignment steps:** Equivalent to the number of steps used in ALIGN-GAP for post-training alignment.

2. **(5e6 + 1e6) total steps:** Adding the steps to get the pretrained agent (5e6) and alignment steps for ALIGN-GAP (1e6 steps) to account for the total steps in ALIGN-GAP.

Experimental results shown in Table 6 reveal that at 1e6 steps, RLPD has not fully converged to achieve effective alignment, as it must simultaneously learn basic environmental behaviors and align with human preferences. ALIGN-GAP, by contrast, achieves alignment more efficiently by leveraging the pre-trained agent's understanding of the environment, allowing it to focus exclusively on preference alignment. Furthermore, even when RLPD is trained for 6e6 steps, including pretraining and alignment, ALIGN-GAP still demonstrates superior alignment performance. This underscores ALIGN-GAP's efficiency and effectiveness in aligning with human preferences on the foundation of a well-trained agent. These results highlight ALIGN-GAP's practical advantages in scenarios where pre-trained agents are available, such as Game AI applications. In practical applications, retraining an agent from scratch can be prohibitively costly; therefore, ALIGN-GAP's approach of achieving alignment through post-training may hold greater practical value.

Table 6: Returns for online trained agent and human performance, and **differences in return to human** for RLPD and ALIGN-GAP. Results are averaged over 5 random seeds.

| Task-Preference | Human Performance | RLPD (1e6 steps) | RLPD (5e6+1e6 steps) | ALIGN-GAP (1e6 steps) |
|---|---|---|---|---|
| Halfcheetah-A | 2.2±0.01 | -1.93±0.41 | -1.35±0.38 | -0.86±0.33 |
| Halfcheetah-B | 125.32±0.76 | -39.22±10.80 | -10.04±3.17 | -0.44±1.97 |
| Halfcheetah-C | 105.27±3.30 | -25.71±2.46 | 4.89±0.15 | -0.91±0.86 |
| Halfcheetah-D | 114.74±0.45 | -13.06±3.25 | -6±1.27 | -7.3±0.29 |
| Walker2d-A | 20.72±2.06 | -5.00±1.79 | -1.84±0.96 | 2.22±3.1 |
| Walker2d-B | 122.11±0.42 | -39.53±9.01 | -14.4±2.03 | -2.87±26.06 |
| Walker2d-C | 116.53±0.74 | -47.01±5.20 | -18.26±5.78 | -1.43±1.78 |
| Walker2d-D | 97.35±26.53 | -32.19±8.62 | 7.74±0.62 | -3.04±27.97 |
| Hopper-A | 30.74±0.01 | -14.8±3.20 | -9.77±0.15 | 6.46±1.56 |
| Hopper-B | 93.88±28.17 | -13.74±6.89 | -12.56±2.30 | -3.33±0.02 |
| Hopper-C | 31.15±4.44 | -18.70±2.71 | -13.06±1.91 | 0.57±2.64 |
| Hopper-D | 110.07±0.35 | -25.01±7.20 | -14.47±3.85 | -8.77±0.49 |

## A.6 COMPUTATIONAL EFFICIENCY ANALYSIS

To evaluate the computational efficiency of ALIGN-GAP, we measured the training speed in iterations per second under different configurations, as shown in Table 7. These results demonstrate that both reward calibration and reward curriculum introduce minimal computational overhead, affirming ALIGN-GAP's suitability for practical applications. The addition of reward calibration introduces a minor computational cost, as it requires an additional forward pass through the reward model for each trajectory to compute calibrated scores. Despite this, the reduction in training speed compared to configurations without calibration is only about 4.4%, highlighting the lightweight nature of this

operation. Similarly, the reward curriculum dynamically adjusts the weighting between environmental rewards and preference-aligned rewards, but this computation is exceedingly efficient, involving only a simple arithmetic operation at each training step. Consequently, the observed reduction in speed with the inclusion of curriculum is only about 0.5%, further demonstrating its computational feasibility.

The results show that ALIGN-GAP's computational cost is nearly identical to configurations omitting one or more of its components. Given the significant alignment improvements observed in our experiments, this slight reduction in training speed represents a reasonable trade-off. Reward calibration and reward curriculum are crucial components for guiding the agent towards human-preferred behaviors while maintaining stability in previously learned skills. Their lightweight design ensures ALIGN-GAP remains efficient even for large-scale training tasks. These findings emphasize the practicality of ALIGN-GAP in scenarios with computational constraints, such as real-world Game AI applications. Furthermore, in settings with severe resource limitations, additional optimization techniques like model quantization, pruning, or parallelized computations can be employed to further reduce computational costs while preserving alignment quality. By prioritizing both alignment performance and efficiency, ALIGN-GAP balances practical applicability with its innovative design.

Table 7: Iteration per second.

| Configuration | Iteration Per Second |
|---|---|
| ALIGN-GAP | 42.61±0.12 |
| ALIGN-GAP w/o Reward Calibration | 44.59±0.09 |
| ALIGN-GAP w/o Reward Curriculum | 42.83±0.11 |

