# OpenReview forum: "Online-to-Offline RL for Agent Alignment"
_ICLR.cc/2025/Conference — ICLR 2025 Poster_

### Official Review · Reviewer_qPd3 · 2024-10-26

**Soundness:** 3
**Presentation:** 3
**Contribution:** 3
**Rating:** 6
**Confidence:** 4

**Summary:**

This paper conceptualizes a new problem setting for reinforcement learning, namely, the online-to-offline setting, which aims to align RL agents with targeted human values in offline data after extensive online training. The proposed method aims at solving challenges in this setting, e.g., agent unlearning caused by rapid reward shifting. Experiements on Game AI verify the effectiveness of the proposed method.

**Strengths:**

- The paper proposes a novel problem setting, namely, online-to-offline RL. The authors clearly explain the distinction between online-to-offline and offline-to-online RL. The practical importance of the newly proposed setting is also well explained.
- The proposed method is simple and intuitive, and it well handles the issue of sharp reward shifting in the online-to-offline setting, which is demonstrated by extensive experiments.

**Weaknesses:**

- The newly conceptualized setting is termed "online-to-offline", however the proposed method for this setting, namely, reward curriculum, needs access to the online environment in the offline phase, which seems inconsistent with the offline concept. Although, considering the object of this new setting, i.e., leveraging potentially insuficient offline data to align the online learned agent, it is somewhat reasonable to assume access to the online environment, this may still limit the application of the proposed method to some real-world scenarios that are truly offline in the second phase.
- The work only focuses on the Game AI context. It would be nice to see if the method works on more practical applications.

**Questions:**

- What causes the agent's failure to learn if we don't explicitly specify "what is preferred by humans but not yet learned by the agent" by Eq. (5)? Since when the preferences of humans and the agent are aligned, $R_{\phi}(\tau)$ would be similar to $R_{\phi}(\tau')$ and so there'll be no significant reward shifting. The motivation to use RM score calibration therefore seems a bit unclear, in contrast to the use of curriculum approach.
- It would be interesting to look into why exactly the agent unlearns when the reward signal shifts rapidly. This reminds me of the recent studies of plasticity loss in the field. I think these work may help understanding this phenomenon.
- Figure 9 is not very persuasive for the purpose of showcasing the effectiveness of the aligning method, since there are only 4 pairs of static images. I think it would be nicer to accompany the paper with videos that compares the performances of different alignment methods.

---

> ### Author Response · Authors · 2024-11-22
>
> Thank you for your constructive review and valuable suggestions! Below, we provide a detailed response to your questions and comments. If any of our responses fail to sufficiently address your concerns, please inform us, and we will promptly follow up.
>
> **[W1] Utilization of online environment during offline alignment in Game AI**
>
> In Game AI applications, a key characteristic is the availability of a game core that facilitates efficient agent training and deployment. During alignment, it is reasonable to assume the continued availability of the game core, as it is typically integral to real-world deployment. For instance, many commercial games rely on persistent game engines to manage online interactions, making such environments accessible during the alignment phase [1,2]. We acknowledge the limitation that this assumption may not hold in scenarios where the online environment is unavailable during alignment. Future work could explore extensions of ALIGN-GAP to fully offline settings, such as using simulated environments or learned models of the environment.
>
> **[W2] Extension to wider domains for agent alignment**
>
> Game AI provides an accessible and evaluable testbed for human alignment methods due to its structured environments and measurable human preferences (e.g., strategies or play styles). Additionally, Game AI remains one of the primary application domains for reinforcement learning agents. That said, ALIGN-GAP is not inherently restricted to Game AI. Its design can be adapted to other domains, such as autonomous driving or healthcare, with appropriate consideration of domain-specific human preferences, their representation, and evaluation. These broader applications are considered as interesting future works and could also address some the domain-specific challenges of defining and aligning with human preferences in more complex real-world scenarios.
>
> **[Q1] Explanation for reward calibration in agent alignment**
>
> Reward calibration (Equation 5) serves to explicitly identify and prioritize behaviors preferred by humans but not yet adopted by the agent. This calibration differentiates between trajectories with high human preference scores ($R_\phi(\tau)$) and those where the agent's actions deviate from human preferences ($R_\phi(\tau')$). When the agent becomes aligned with human preferences, $R_\phi(\tau)$ and $R_\phi(\tau')$ converge, resulting in negligible reward shifts and halting further training—exactly as intended. This mechanism ensures that calibration effectively guides the agent during misalignment and stops adjustments once alignment is achieved, complementing the curriculum approach.
>
> **[Q2] Analysis on the unlearning phenomenon when shifting between rewards**
>
> We agree that rapid shifts in reward signals can destabilize agents and cause unlearning. This phenomenon is explored in Appendix A.2, where we analyze the EPIC distance [3] between the environmental and alignment rewards. The EPIC distance is a metric designed to directly quantify the difference between two reward functions. By designing a reward curriculum, we demonstrate that the EPIC gap can be reduced to $\frac{2}{3}$ of its original value, thereby smoothing the transition and mitigating unlearning. We appreciate your suggestion regarding studies on plasticity loss and will incorporate insights from this field to deepen our understanding of reward signal switching [4,5]. This will guide future refinements to ALIGN-GAP’s design and theoretical analysis, and any new results will be further updated in the final version.
>
> **[Q3] Visualization on the alignment results**
>
> Using videos indeed provides a more intuitive demonstration of the agent's alignment with human preferences. To supplement the results shown in Figure 9, we have created corresponding visualizations and uploaded them to an anonymous GitHub repository (https://anonymous.4open.science/r/test2-08C1/video.mp4) to showcase ALIGN-GAP's effectiveness in aligning agent behavior with human preferences. In the final version, we will include a more comprehensive collection of video comparisons across additional environments and a broader range of human preferences. This will offer a clearer and more persuasive demonstration of ALIGN-GAP's performance and its ability to align agents with diverse human behaviors. Thank you for your valuable suggestion.
>
> [1] Serious games in digital gaming: A comprehensive review of applications, game engines and advancements.
>
> [2] Game engine architecture.
>
> [3] Quantifying Differences in Reward Functions.
>
> [4] A Study of Plasticity Loss in On-Policy Deep Reinforcement Learning.
>
> [5] Loss of Plasticity in Continual Deep Reinforcement Learning.

---

> > ### Comment · Reviewer_qPd3 · 2024-11-25
> >
> > Thank you for the response and clarification. The video is intuitive for comparisons of performance. Most of my concerns are addressed and thus I have accordingly raised my confidence to 4.

---

### Official Review · Reviewer_AHtP · 2024-10-27

**Soundness:** 3
**Presentation:** 3
**Contribution:** 3
**Rating:** 6
**Confidence:** 3

**Summary:**

The paper  introduces ALIGN-GAP, an innovative framework that enables alignment of high-performing RL agents trained online with specific human preferences. This framework consists of two main components: a reward model trained on limited offline human data to capture preferences and a calibrated curriculum-based learning approach to gradually adjust the agent's goals from maximizing environmental rewards to human-aligned behaviors.
Through experiments in diverse game environments, the ALIGN-GAP approach demonstrates effective alignment across various human styles, achieving behavior that closely resembles human preferences.

**Strengths:**

1) **Significance of the Problem**: The paper addresses a crucial and emerging issue in reinforcement learning — aligning agent behavior with human preferences, especially in cases where human data is limited and offline. By tackling the alignment problem in online-to-offline RL contexts, this work directly addresses gaps in current RL applications and sets a foundation for broader, practical deployment of RL agents in human-interactive settings.

2) **Well-structured and Methodologically Clear**: The paper is logically structured, with each section building effectively on the last. The clear delineation between problem formulation, related work, methodology, and experimental results enhances readability and comprehension, making it easy to follow the progression from theoretical motivation to practical implementation.

3) **Robust Experimental Results**: The experimental results demonstrate the effectiveness of ALIGN-GAP across a variety of environments and human preference types, showcasing the method’s versatility. The use of both qualitative and quantitative evaluations provides a compelling case for the approach, with detailed analysis that underscores the alignment improvements made by ALIGN-GAP over baselines.

4) **Rationale Behind the Proposed Method**: The approach is intuitively sensible, leveraging a transformer-based reward model and calibrated curriculum learning in a way that aligns well with recent advancements in human-preference modeling. The stepwise, calibrated shift from performance-based to human-aligned behaviors ensures that the agent maintains learned skills while adapting effectively, reinforcing the practical value and feasibility of the proposed solution.

**Weaknesses:**

1) **Preference Learning Paradigm Clarity**: In Section 4.1, the method constructs a preference learning framework where human trajectories are set as the “winner” samples and agent trajectories as the “loser” samples. This choice is somewhat unclear because agent trajectories may still exhibit successful behavior, albeit not fully aligned with human preferences. This setup raises concerns about whether such binary labeling could unintentionally degrade or "unlearn" the agent's effective skills. It would be beneficial if the authors could further clarify how the method differentiates between genuinely preferred human behavior and successful agent behavior to avoid compromising the agent's performance.

2) **Impact of Agent Pretraining**: Building on the previous point, it remains unclear how essential the agent pretraining phase is to the final performance. Would the method still be effective if the agent was trained exclusively with the human preference-aligned reward model, without prior environmental training? Additionally, an analysis of how different levels of pretraining influence the results would provide insight into the dependence on the agent's initial training stage and could help clarify the method’s requirements.

3) **Potential Time and Computational Costs**: The ALIGN-GAP approach may be computationally intensive due to the need for separate alignment processes for each human player’s unique preferences. This potential cost in terms of training time, especially for applications requiring frequent preference realignments, raises questions about scalability. It would be valuable for the authors to discuss potential optimizations, such as reusing alignment data across similar preferences or developing faster adaptation techniques to make this approach more time-efficient for practical use.

**Questions:**

Please see weaknesses.

---

> ### Author Response · Authors · 2024-11-22
>
> Thank you for your constructive review and valuable suggestions! Below, we provide a detailed response to your questions and comments. If any of our responses fail to sufficiently address your concerns, please inform us, and we will promptly follow up.
>
> **[W1] Clarification on preference learning for agent alignment**
>
> In Section 4.1, we use human trajectories as "winner" samples and agent trajectories as "loser" samples to train the reward model. This binary labeling framework is designed to encode human preferences, a common approach in preference learning literature [1,2]. This setup allows us to encode human preferences effectively, but we acknowledge the concern that some agent trajectories, while not perfectly aligned with human preferences, may still represent successful behavior that should not be penalized.
>
> To address this, ALIGN-GAP employs reward calibration (Equation 5) to distinguish between human-preferred behaviors and agent-learned behaviors. Calibration compares the reward scores of human-preferred and agent trajectories under the same states. When the agent's behavior aligns with human preferences, the scores converge ($R_\phi(\tau) \approx R_\phi(\tau')$), indicating no need for adjustment. In contrast, for misaligned behaviors, calibration amplifies the difference in scores, prioritizing areas that require improvement while preserving effective behaviors. Furthermore, the reward curriculum (Equation 6) ensures that the alignment process proceeds gradually. Early in training, environmental rewards dominate, which helps retain the agent's existing skills, even those not fully aligned with human preferences. As training progresses, the focus shifts towards aligning with human preferences, encouraging the agent to refine and adapt its behaviors. This gradual transition helps prevent unlearning during the early phases of alignment while progressively emphasizing human-aligned behaviors.
>
> **[W2] Impact of pretraining for agent alignment**
>
> Our design choice of aligning pre-trained agents aligns with real-world Game AI applications, where substantial resources are often already invested in training high-performing agents [3, 4]. Fine-tuning these agents for human preference alignment serves as a post-training iterative step, ensuring that the alignment process builds upon the agent’s foundational skills (similar to LLM pretraining and post-training alignment). In contrast, training a new agent from scratch involves learning both the environmental dynamics and the alignment simultaneously. This dual learning process could result in slower convergence and higher computational demands compared to fine-tuning a pretrained agent that has already mastered the environmental tasks. As an experimental support, we update a comparison in Appendix A.5 between ALIGN-GAP and an RLPD-like online alignment with offline data [5], without prior environment pretraining. Our results show that with equivalent training steps, ALIGN-GAP achieves superior alignment performance. Even when extending the training duration of the RLPD-like approach to match the combined steps of pretraining and ALIGN-GAP fine-tuning, the alignment outcomes remain inferior to ALIGN-GAP. Besides, in the final version, we will include additional ablation studies to systematically analyze the impact of agent pretraining levels on alignment outcomes.
>
> **[W3] Computational cost for components in ALIGN-GAP**
>
> The additional computational cost of ALIGN-GAP is minimal relative to the overall training process. Reward calibration involves lightweight operations such as forward passes through the reward model, and curriculum learning primarily entails reweighting the reward function, introducing negligible overhead. In our updated Appendix A.6, we compared the number of iterations per second achievable with and without reward calibration or curriculum learning under the same hardware conditions. The results show that incorporating these components does not introduce significant additional computational overhead. To enhance scalability, we could benefit from techniques like model quantization and pruning, which can optimize the reward model for inference during alignment (considering the twice inference in reward calibration) [6,7]. Additionally, grouping similar human preferences through clustering could allow for reusable alignment processes, reducing the need for frequent retraining.
>
>
>
> [1] Direct Preference Optimization: Your Language Model is Secretly a Reward Model.
>
> [2] Provable Offline Preference-Based Reinforcement Learning.
>
> [3] A Survey on Game Playing Agents and Large Models: Methods, Applications, and Challenges.
>
> [4] Pre-trained Language Models as Prior Knowledge for Playing Text-based Games.
>
> [5] Efficient Online Reinforcement Learning with Offline Data.
>
> [6] Towards Efficient Post-training Quantization of Pre-trained Language Models.
>
> [7] SmoothQuant: Accurate and Efficient Post-Training Quantization for Large Language Models.

---

> ### Author Response · Authors · 2024-12-01
> **End of Discussion Period**
>
> Dear Reviewer AHtP,
>
> We appreciate all of the valuable time and effort you have spent reviewing our paper. As the discussion period is ending, we gently request that you review our reply and consider updating your evaluation accordingly. We believe that we have addressed all questions and concerns raised, but please feel free to ask any clarifying questions you might have before the end of the discussion period.
>
> Best,
>
> Authors

---

### Official Review · Reviewer_rThM · 2024-11-03

**Soundness:** 2
**Presentation:** 3
**Contribution:** 2
**Rating:** 6
**Confidence:** 4

**Summary:**

The paper is well-written and clearly presents ALIGN-GAP, a simple method for aligning reinforcement learning (RL) agents with human preferences through an online2offline framework. ALIGN-GAP leverages a Transformer-based reward model to extract human preferences from limited offline data and utilizes a curriculum-based preference learning strategy to gradually shift the agent’s objectives from maximizing environmental rewards to aligning with human values. Extensive experiments across D4RL locomotion tasks and Atari games demonstrate that ALIGN-GAP effectively aligns agent behaviors with human preferences, outperforming several baseline methods. The thorough evaluations, including ablation studies, highlight the robustness and generalizability of the approach. However, the method lacks of theoretical foundations and also suffers from a limited range of experiments. While it performs well in the selected D4RL and Atari game environments, the scope of tasks is relatively narrow.

**Strengths:**

The strengths of the paper include its clear and well-organized writing, which makes the proposed approach easy to follow. Additionally, the method itself is intuitive and relatively simple, combining straightforward components like a reward model and curriculum learning for preference alignment. This simplicity makes the approach accessible and potentially easier to implement, enhancing its appeal for practical applications.

**Weaknesses:**

The method lacks of theoretical foundations and also suffers from a limited range of experiments. While it performs well in the selected D4RL and Atari game environments, the scope of tasks is relatively narrow.

**Questions:**

1. **Data Requirements**: The method relies on high-quality offline human data to build a preference model. How sensitive is ALIGN-GAP to the quality and quantity of this data? Would noisy or sparse data significantly impact the model’s performance?
2. **Theoretical Guarantees**: The paper lacks a rigorous theoretical analysis. Are there any formal guarantees on the stability or convergence of the agent's alignment process, especially when switching between environmental and human-aligned rewards?
3. **Computational Cost and Efficiency**: Given the multi-component nature of ALIGN-GAP (e.g., reward calibration, curriculum learning), what is the computational cost of this method compared to simpler baselines? Could this limit its feasibility for resource-constrained applications?
4. **Evaluation Metrics**: The paper evaluates alignment through performance similarity to human proxies. However, this measure is somewhat subjective. Are there more objective or standardized metrics that could improve the evaluation of alignment quality?

---

> ### Author Response · Authors · 2024-11-22
>
> Thank you for your constructive review and valuable suggestions! Below, we provide a detailed response to your questions and comments. If any of our responses fail to sufficiently address your concerns, please inform us, and we will promptly follow up.
>
> **[W1,Q2] Range of experimental environments and theoretical analysis for switching between different rewards**
>
> We evaluates ALIGN-GAP on two widely-used reinforcement learning benchmarks: D4RL locomotion tasks (e.g., HalfCheetah, Walker2D, and Hopper) and Atari games (e.g., Pac-Man and Space Invaders). These environments were chosen due to their prevalence in prior research and their diverse characteristics, allowing us to test ALIGN-GAP across a range of alignment challenges [1,2]. To further strengthen our argument, we have now extended ALIGN-GAP on the Alien game, as detailed in Appendix A.3. In future work, we aim to explore alignment in more complex, application-oriented game scenarios with real-time human interactions to expand the scope of our method. We will update the observations in the final version and future iterations of our work to broaden the applicability of ALIGN-GAP.
>
> In Appendix A.2, we present an analysis of the impact of our curriculum learning strategy on alignment. Specifically, we formally demonstrate that introducing a curriculum reduces the expected Equivalent-Policy Invariant Comparison (EPIC) distance [3] between the environmental reward and the alignment objective to $\frac{2}{3}$ of their original distance. The EPIC distance is a metric designed to directly quantify the difference between two reward functions without requiring a policy optimization step. It is proven to be invariant within an equivalence class of reward functions that induce the same optimal policy, making it a robust measure for this purpose. This reduction in EPIC distance supports the theoretical stability of switching between environmental and human-aligned rewards and highlights the effectiveness of the curriculum in guiding alignment.
>
> **[Q1] Data requirements for alignment with ALIGN-GAP**
>
> ALIGN-GAP is designed to work effectively with small amounts of human preference data, as motivated in Section 1, addressing practical challenges of data collection costs and availability. For example, in Section 5.1, we show that ALIGN-GAP achieves effective alignment with as few as 10 episodes of human data. This efficiency is attributed to the reward model's ability to encode human preferences even from sparse data. Regarding noisy or low-quality data, ALIGN-GAP's reliance on pairwise comparisons in training the reward model provides inherent robustness (Section 5.4). However, excessively noisy or poorly representative data could degrade alignment performance, as with any preference-based RL methods.
>
> **[Q3] Computational cost and efficiency for components in ALIGN-GAP**
>
> The additional computational cost of ALIGN-GAP is minimal compared to the overall training process. Key components such as reward calibration and curriculum learning primarily involve reward computations, which add negligible overhead relative to the neural network training for the agent. As shown in Appendix A.6, we compared the number of iterations per second achievable with and without reward calibration or curriculum learning under the same hardware conditions. The results show that incorporating these components does not introduce significant additional computational overhead. For resource-constrained applications, techniques such as model quantization and pruning can further reduce the computational burden of the reward model, making ALIGN-GAP practical and efficient even in such scenarios [4,5].
>
> **[Q4] Evaluation metrics for degree of alignment**
>
> We agree that alignment evaluation inherently involves some subjectivity due to the nature of human preferences. Currently, we assess alignment by measuring performance similarity to human proxies (Table 1 and 2) and visualizing agent behavior styles (Figures 5–9). These methods align with standard practices in LLM and agent alignment research, which often rely on reward model scores or human evaluations (e.g., win rates, subjective ratings) [6]. Besides, we have created corresponding visualizations for the observations in Figure 9 (https://anonymous.4open.science/r/test2-08C1/video.mp4) for a clearer presentation. To enhance evaluation robustness, we will explore combining subjective measures with more objective metrics in future work.
>
> [1] Offline Reinforcement Learning with Implicit Q-Learning.
>
> [2] A Review for Deep Reinforcement Learning in Atari:Benchmarks, Challenges, and Solutions.
>
> [3] Quantifying Differences in Reward Functions.
>
> [4] Towards Efficient Post-training Quantization of Pre-trained Language Models.
>
> [5] SmoothQuant: Accurate and Efficient Post-Training Quantization for Large Language Models.
>
> [6] Trustworthy LLMs: a Survey and Guideline for Evaluating Large Language Models' Alignment.

---

> > ### Comment · Reviewer_rThM · 2024-11-26
> >
> > My most of my questions have been answered, so I will increase the rating to 6.

---

### Official Review · Reviewer_9gAS · 2024-11-04

**Soundness:** 2
**Presentation:** 3
**Contribution:** 3
**Rating:** 5
**Confidence:** 4

**Summary:**

The paper proposes an online-to-offline reinforcement learning (RL) framework, ALIGN-GAP, aimed at aligning RL agents’ behavior with human preferences, specifically within Game AI environments. This framework incorporates a transformer-based reward model to encode human preferences from limited offline data and a calibrated preference curriculum that incrementally transitions the agent's learning objectives from maximizing environmental rewards to prioritizing human-aligned behaviors. The experimental results across various game environments highlight ALIGN-GAP’s effectiveness in reducing the alignment gap between agents and humans.

**Strengths:**

- The approach of fine-tuning a well-trained agent to match human preferences is novel and previously unexplored. The paper presents various challenges for this process and achieves strong performance through well-designed reward calibration and curriculum learning.

- Provides a theoretical justification for the reward calibration method, lending a solid foundation to the alignment strategy.

- Through comprehensive testing, ALIGN-GAP shows consistent improvement in agent-human alignment, proving its practicality and effectiveness in real-world scenarios like gaming.

**Weaknesses:**

- I'm curious if fine-tuning a well-trained agent is necessary for aligning it with human preferences. Could training an agent offline, like AlignDiff [1], using the rollout and replay buffer data from the well-trained agent, be more advantageous?
Here’s my perspective in more detail:

  - Fine-tuning a well-trained agent to align with human preferences is challenging, particularly due to unlearning risks. While techniques the authors proposed like reward calibration and curriculum learning help, they may not fully solve the problem. Research (e.g., RLPD [2]) suggests that training a new agent using pre-collected data is often more stable and effective than fine-tuning.
  - Thus, instead of direct fine-tuning, using the well-trained agent’s replay buffer and rollout data to train a new, preference-aligned policy—either offline or online with offline data settings like RLPD—could be more effective.
  - The paper criticizes AlignDiff for needing manually designed attributes, but this seems inconsistent. For example, the MuJoCo experiments also rely on manually crafted attributes (such as health, movement, and cost), similar to AlignDiff. In Atari tasks, the paper relies on UMAP for unsupervised clustering of important attributes, but I think manually designing attributes might be simpler and more intuitive compared to relying on additional UMAP training.

  Therefore, it would be helpful if the authors could explain the specific benefits of the proposed method compared to using AlignDiff’s offline training (or an RLPD-like setting), supported by experimental results.

- Although the introduction of this paper suggests a focus on game AI settings, the experiments were primarily conducted on MuJoCo locomotion tasks and two Atari games. Expanding the range of experiments to include more varied game environments could help build a more cohesive argument.

[1] Dong, Zibin, et al. "AlignDiff: Aligning Diverse Human Preferences via Behavior-Customisable Diffusion Model." The Twelfth International Conference on Learning Representations.

[2] Ball, Philip J., et al. "Efficient online reinforcement learning with offline data." International Conference on Machine Learning. PMLR, 2023.

**Questions:**

- In Section 5.4, $\lambda$ in the expression for $\alpha(t)$ appears to be an important hyperparameter, but I couldn't find any information in the paper on how to tune $\lambda$. Could you explain how you tuned $\lambda$ for the experiments with the Linear / Exponential \$\alpha(t)$?
- In equation (5), it seems that $R_{\text{calibrated}}$ might need to be scaled differently depending on the environment. Does it empirically perform well with just simple subtraction, without considering such adjustments?

---

> ### Author Response · Authors · 2024-11-22
>
> Thank you for your constructive review and valuable suggestions! Below, we provide a detailed response to your questions and comments. If any of our responses fail to sufficiently address your concerns, please inform us, and we will promptly follow up.
>
> **[W1] Justification for fine-tuning Instead of re-training, and comparison with AlignDiff**
>
> 1. Justification for Fine-tuning Instead of Training a New Agent: Our work addresses the challenge of aligning well-trained agents with human preferences as a post-training iterative step, which assumes the existence of a pre-trained policy. This approach reflects the practical realities of real-world Game AI applications, where significant computational and resource investments are made to develop high-performing agents [1,2]. Fine-tuning these agents for alignment leverages their existing understanding of the environment, enabling efficient adaptation to human preferences. Re-training an agent from scratch like RLPD may suffer from slower convergence and longer training times compared to fine-tuning an existing agent that has already mastered environmental dynamics. To validate this claim, we update a comparison in Appendix A.5 between ALIGN-GAP and an RLPD-like online alignment with offline data settings. Our results show that with equivalent training steps, ALIGN-GAP achieves superior alignment performance. Even when extending the training duration of the RLPD-like approach to match the combined steps of pretraining and ALIGN-GAP alignment, its performance remain inferior to ALIGN-GAP.
>
> 2. Comparison to AlignDiff and Attribute Design: In AlignDiff, attributes are explicitly used as language commands to train both the attribute model and the diffusion model. In contrast, we employ data clustering techniques (e.g., using reward oracles or UMAP) to group human preference data by styles during preprocessing. As described in Section 4, ALIGN-GAP requires only preference data with specific styles for alignment and does not rely on additional attribute information. The use of reward oracles or UMAP is solely for preprocessing the data. This design makes ALIGN-GAP more flexible and better suited for aligning agents with specific user groups or individual preferences, as it avoids reliance on domain-specific attribute definitions. In applications, if a game designer aims to align an agent with the play style of a particular user group or even a single player, ALIGN-GAP can use the collected data from that user as human preference data without additional attribute designs.
>
> **[W2] Expand the range of experiments to include more game environments**
>
> We agree that diverse environments are important to demonstrate generality. While our experiments primarily focus on MuJoCo locomotion tasks and Atari games, these are widely used benchmarks in reinforcement learning research for Game AI due to their well-defined task dynamics and popularity in prior studies [3,4]. To address the concern, we have extended our experimental evaluation to include the Alien game to examine the alignment capability of ALIGN-GAP, shown Appendix A.3. In future work, we aim to extend ALIGN-GAP's applicability by exploring alignment in more complex, real-time strategy games with real-time human interactions. We will update the observations in the final version and future iterations of our work to broaden the applicability of ALIGN-GAP.
>
> **[Q1] Explanation on $\lambda$**
>
> In our experiments, $\lambda$ is set to control the transition of $\alpha(t)$ from 0 (environment reward focus) to 1 (reward model focus) as curriculum learning progresses. Specifically:
> - Linear $\alpha(t)$: As detailed in Section 4.2, $\alpha(t) = \frac{\text{current\ step}}{\text{total\ steps}}$ ensures a smooth, linear increase.
> - Exponential $\alpha(t)$: We empirically set $\lambda = 5$, which is a positive constant to ensure that $\alpha(t)$ follows the trend of increasing from 0 to approaching 1.
> We have updated the setting for $\lambda$ in the revised version for clarity.
>
> **[Q2] The scale of $R_\text{calibrated}$**
>
> Yes, empirical results demonstrate that $R_{\text{calibrated}}$ obtained via subtraction performs robustly across environments without additional scaling. The reward model's score ranges, observed to be approximately [-8, 8] across tasks (e.g., Walker2D in Figure 4(b)), remain consistent in different environments, which aligns with findings in related reward model alignment studies [5]. We acknowledge that exploring environment-specific scaling RM scores adaptively could enhance performance in future work.
>
> [1] A Survey on Game Playing Agents and Large Models: Methods, Applications, and Challenges.
>
> [2] Pre-trained Language Models as Prior Knowledge for Playing Text-based Games.
>
> [3] Offline Reinforcement Learning with Implicit Q-Learning.
>
> [4] A Review for Deep Reinforcement Learning in Atari:Benchmarks, Challenges, and Solutions.
>
> [5] Secrets of RLHF in Large Language Models Part II: Reward Modeling

---

> ### Comment · Reviewer_9gAS · 2024-11-27
>
> I deeply appreciate the author’s effort to address my question and conduct additional experiments.
>
> **[finetuning vs retraining]**
>
> I apologize for not explaining clearly earlier, which might have caused some misunderstandings. "RLPD-like" was just one example I mentioned, but what I was more curious about was what happens if, in addition to using an RLPD-like method, we train from scratch offline using AlignDiff. (I understand the differences between the proposed method and the "RLPD-like" method. Thank you for the response.)
>
> As I understand it, the current method involves training a reward model by combining agent trajectories generated by the online-trained agent with offline human trajectories. This reward model is then used to fine-tune the online-trained agent. However, my thought was: why not use the combined data of the same agent trajectories and offline human trajectories to train a new agent with freshly initialized weights directly using AlignDiff? Alternatively, if offline human trajectories are available, wouldn't it suffice to simply train AlignDiff on those?
>
> I agree with the author’s comment that training from scratch is naturally more costly than fine-tuning, especially in environments like game applications. However, if fine-tuning causes issues such as an irrecoverable performance drop, as discussed in many existing offline-to-online RL studies, then training from scratch might be a stable alternative worth considering despite the cost. Furthermore, while the papers cited by the author deal with truly costly domains, I don’t believe the MuJoCo or Atari experiments presented in this paper represent environments where the tradeoff would definitively justify the conclusion that fine-tuning is always preferable.
>
> I would like to understand the differences between and the pros and cons of the following approaches:
>
> (1) RL agent training → Fine-tuning the RL agent using its rollout data + human preference data
>
> (2) RL agent training → Training a new agent using AlignDiff (with reward oracle or UMAP attributes) on the agent's rollout data + human preference data
>
> (3) Training AlignDiff (with reward oracle or UMAP attributes) using only human preference data
>
> &nbsp;
>
> **[methods for defining attributes]**
>
> Additionally, in MuJoCo, the environment preference reward is defined based on three dimensions: maintaining health, moving forward, and controlling cost. It would be ideal to extract and use reward oracles solely from offline data without domain knowledge. However, if that’s not feasible, it seems this would also require domain knowledge to define, making it essentially similar to defining attributes in AlignDiff. I now clearly understand the differences in attribute definitions between AlignDiff with utilizing UMAP.
>
> &nbsp;
>
> I find the problem of aligning trained agents with human preferences both important and fascinating. However, the proposed method seems somewhat intricate in its design to address issues from online fine-tuning, which makes me wonder whether this complexity is necessary or if alternative approaches might be possible. While offline and online fine-tuning can be seen as fundamentally different, I feel that a more comprehensive comparison with prior works focused on aligning agents with human preferences would be valuable.

---

> > ### Author Response · Authors · 2024-11-29
> >
> > Thank you for your constructive feedback. Below, we address your questions regarding fine-tuning vs. retraining and attribute design for alignment. We hope these clarifications could address your concerns and strengthen the case for our approach.
> >
> > **Fine-tuning vs. Retraining, and Attribute Design for Alignment**
> >
> > We appreciate your thoughtful question regarding the trade-off between fine-tuning a well-trained agent and training a new agent from scratch, as seen in methods like AlignDiff. This decision depends on several factors, including performance, training costs, and the cost of developing and labeling attributes.
> >
> > In Method (1) (fine-tuning with rollout and human preference data, as shown by the baseline "Finetune w/SAC" in Table 1), the agent is adapted to human preferences. However, when human data significantly diverges from the agent's prior training (e.g., when it represents novice or less skilled behavior), the agent may continue to prioritize previously learned behaviors, as its training objective remains maximizing environmental returns, rather than adapting to the new human preferences.
> >
> > For Method (2) and Method (3), both approaches using AlignDiff require careful design of attributes to train the attribute model. Since AlignDiff uses more detailed attributes (e.g., "Torso height" and "Stride length" in the Walker environment) rather than simply relying on reward dimensions like in MuJoCo, this involves labeling the agent's rollout data (in Method (2)) and human preference data (in both Method (2) and Method (3)) through crowdsourcing (as done in AlignDiff), which can be time-consuming and expensive in practice.  This level of detail may help improve the model's ability to capture and align specific human preferences, but it also incurs additional data labeling costs. In contrast, ALIGN-GAP could use multiple ways to process human preference data such as reward oracles or unsupervised clustering techniques like UMAP, making it more flexible and reducing the reliance on explicit attribute design. This flexibility could allow ALIGN-GAP to adapt across various environments, even when attribute labeling is costly or even impractical.
> >
> > In practical applications, both defining preferences via domain knowledge and using unsupervised techniques have their value. Domain knowledge helps leverage insights specific to a game, while unsupervised techniques enable the direct use of data from specific player groups without the need for extensive annotation. ALIGN-GAP can accommodate both approaches, making it adaptable to a wide range of use cases. We plan to further explore the impact of different human preference data sources (e.g., UMAP clustering in MuJoCo) and will compare these results with AlignDiff, including experiments where AlignDiff is trained using MuJoCo reward attributes.
> >
> >
> > **Exploring Alternative Approaches to Agent-Human Alignment**
> >
> > We agree that the problem of agent-human alignment is an exciting and challenging area of research. As demonstrated in Section 5.4, our ablation study validates the necessity of components like reward calibration and preference curriculum in ALIGN-GAP. However, we also recognize that alternative approaches for alignment exist, and we plan to explore them further in future work.
> >
> > In the final version of this paper, we will include a more comprehensive comparison with AlignDiff, especially considering alternative attribute designs such as MuJoCo reward attributes. Additionally, we will continue to explore improved alignment methods for pre-trained agents from wider perspectives, such as value estimation errors, as discussed in offline RL and offline-to-online RL literature [1,2,3]. These analyses will help refine our understanding of alignment techniques and further validate the effectiveness of ALIGN-GAP. Thank you again for your insightful and constructive feedback.
> >
> > [1] Conservative Q-Learning for Offline Reinforcement Learning.
> >
> > [2] Offline Reinforcement Learning with Implicit Q-Learning.
> >
> > [3] Cal-QL: Calibrated Offline RL Pre-Training for Efficient Online Fine-Tuning.

---

> > > ### Comment · Reviewer_9gAS · 2024-11-29
> > >
> > > Thank you so much for your thoughtful response. However, I feel that it slightly misses some of my key points and doesn’t fully address them.
> > >
> > > I see differences between AlignDiff and ALIGN-GAP in (a) the method of attribute design and (b) the training approach—whether a new agent is trained from scratch using rollout data or a pre-trained agent is fine-tuned. In my earlier comparison of (1), (2), and (3), I specifically asked about the comparison involving "reward oracle or UMAP attributes," i.e., AlignDiff using ALIGN-GAP's attribute design method. My intention was to better understand the strengths and weaknesses that might arise when (a) remains consistent but (b) differs. Assuming the attribute design is the same, I wanted to explore whether leveraging AlignDiff, which is designed to align well with these attributes, and training it with well-trained agent rollout data and human preference data might present some advantages. As I mentioned earlier, while training from scratch could seem inefficient, it might help mitigate issues like distribution shift, as highlighted in offline-to-online research, potentially offering certain benefits. That was the focus of my question.
> > >
> > > Regarding the difference in (a), as I’ve mentioned before, I still don’t fully understand why AlignDiff’s attribute design method is seen as inferior to ALIGN-GAP’s. For example:
> > >
> > > - AlignDiff uses attributes like "Speed," "Torso height," "Stride length," "Left-right leg preference," and "Humanness," which appear to be defined based on domain knowledge.
> > > - ALIGN-GAP uses attributes like "maintaining health," "moving forward," and "controlling cost," which are defined using a reward oracle.
> > >
> > > But isn’t the reward oracle also based on domain knowledge? If domain knowledge is a factor in both cases, the two methods seem quite similar to me.
> > >
> > > I am not suggesting that the problem ALIGN-GAP addresses is unimportant. Rather, I was hoping for a clearer and more persuasive explanation of whether alternative, potentially simpler approaches leveraging prior research might address the problem effectively. Such an explanation would better highlight the importance of ALIGN-GAP’s methodology and its potential to make a significant contribution to the community. For now, I have decided to maintain my current score.

---

> > > > ### Author Response · Authors · 2024-12-01
> > > >
> > > > Thank you for your thoughtful feedback! To better address your concern about the differences between ALIGN-GAP and related approaches like AlignDiff, we conducted experiments comparing the two methods while keeping (a) the method of attribute design consistent (using attributes from the reward oracle:  "maintaining health," "moving forward," and "controlling cost") and varying (b) the training approach. Specifically, ALIGN-GAP fine-tunes an online-trained agent using both game environment and human preference data, while AlignDiff re-trains the agent using labeled rollout data and human preference data, or only human preference data. We used the same performance metric as in Table 1 of the paper (the performance difference from human, where smaller absolute values indicate better alignment), and evaluated the agent's alignment in the Halfcheetah environment under various human preferences (as defined in Section 5.1). The results are as follows:
> > > >
> > > > | Task-Preference | ALIGN-GAP  | AlignDiff (w/ human data) | AlignDiff (w/ rollout data and human data) |
> > > > | --------------- | ---------- | ---------------------- | ------------------------------------- |
> > > > | Halfcheetah-A   | -0.86±0.33 | 10.85±0.21             | 6.53±0.16                            |
> > > > | Halfcheetah-B   | -0.44±1.97 | 8.97±2.20              | 8.02±2.38                             |
> > > > | Halfcheetah-C   | -0.97±0.86 | 6.13±1.74              | 4.10±1.50                             |
> > > > | Halfcheetah-D   | -7.3±0.29  | -12.51±0.44            | -10.72±0.27                           |
> > > >
> > > > As shown in the table, ALIGN-GAP and AlignDiff both achieve alignment to some degree, but ALIGN-GAP consistently performs better across different human preferences. Notably, AlignDiff shows improved alignment when additional labeled rollout data is included. This improvement is likely due to the increased coverage and diversity of the data, which helps the agent learn more robust policies. However, ALIGN-GAP still outperforms AlignDiff in aligning with human preferences across various types of preferences. We believe this is due to ALIGN-GAP's ability to continue leveraging interactions with the environment during fine-tuning, allowing it to explore and align strategies with human preferences in the game (as detailed in Section 4.2 and Figure 2). In contrast, AlignDiff relies solely on offline training. This aligns with findings in both offline and online RL literature, which emphasize the benefits of environmental exploration during training [1,2].
> > > >
> > > > In practice, ALIGN-GAP could offer greater flexibility in utilizing preference data. It can process human preference data either through manually defined attributes or using unsupervised methods like UMAP. In future experiments, we plan to explore more cost-effective preference data partitioning techniques and design improved methods for agent-human alignment. We hope that ALIGN-GAP will provide an efficient fine-tuning process that enhances RL agents' ability to learn and adapt when aligning with human preferences.
> > > >
> > > > [1] Policy Expansion for Bridging Offline-to-Online Reinforcement Learning.
> > > >
> > > > [2] Cal-QL: Calibrated Offline RL Pre-Training for Efficient Online Fine-Tuning.

---

### Author Response · Authors · 2024-11-22
**Summary of Updates in the Revised Version**

We thank all reviewers for their valuable feedback and constructive suggestions, which have greatly helped us improve the quality of our work. Below, we summarize the main changes and additions in the updated version of our paper, addressing the points raised during the review process:

1. **Comparison with RLPD-like alignment without pretraining:**
We have added a detailed comparison with RLPD-like alignment approaches that do not utilize a pre-trained agent. In Appendix A.5, we demonstrate that ALIGN-GAP, by leveraging a well-trained agent for post-training alignment, achieves superior alignment efficiency and effectiveness compared to re-training from scratch. This supports ALIGN-GAP's focus on Game AI applications where pre-trained agents are commonly available and cost-efficient alignment is essential.

2. **Expanded experimental validation:**
While our main experiments focus on commonly used Game AI benchmarks (D4RL locomotion tasks and Atari games), we have extended the Atari experiments to include the Alien game, showcasing consistent advantages of ALIGN-GAP in Appendix A.3. The findings confirm ALIGN-GAP's consistent performance advantages across different environments. We also outline future plans to test ALIGN-GAP in more complex, application-oriented game scenarios.

3. **Computational cost analysis:**
In Appendix A.6, we provide a comparison of the computational overhead introduced by ALIGN-GAP components, such as reward calibration and reward curriculum. The result shows that these components add minimal overhead relative to the overall training process, ensuring ALIGN-GAP remains computationally efficient and feasible for practical applications.

4. **Enhanced visualization for alignment:**
To provide a more intuitive understanding of ALIGN-GAP's alignment performance, we have created videos corresponding to Figure 9 (shown in https://anonymous.4open.science/r/test2-08C1/video.mp4), showcasing the alignment of agents to diverse human preferences. In future revisions, we plan to include a comprehensive collection of video comparisons across additional environments and a broader range of human preferences.

---

### Meta-Review · Area_Chair_HXra · 2024-12-19

**Metareview:**

The reviewers agree this work presents a novel and intuitive approach for fine-tuning pretrained game agents to align with offline human preferences. In particular, the reward curriculum is insightful and the experiments demonstrate that this approach makes a significant impact.

**Additional Comments On Reviewer Discussion:**

Reviewers mainly asked for additional methodological details and a few reviewers requested results on additional environments. The author addressed these concerns in their rebuttal.

---

### Decision · Program_Chairs · 2025-01-22

Accept (Poster)